

# Locally quasi-stationary states in noninteracting spin chains

**Maurizio Fagotti**[*]

Université Paris-Saclay, CNRS, LPTMS, 91405, Orsay, France

[*] maurizio.fagotti@universite-paris-saclay.fr

## Abstract

Locally quasi-stationary states (LQSS) were introduced as inhomogeneous generalisations of stationary states in integrable systems. Roughly speaking, LQSSs look like stationary states, but only locally. Despite their key role in hydrodynamic descriptions, an unambiguous definition of LQSSs was not given. By solving the dynamics in inhomogeneous noninteracting spin chains, we identify the set of LQSSs as a subspace that is invariant under time evolution, and we explicitly construct the latter in a generalised XY model. As a by-product, we exhibit an exact generalised hydrodynamic theory (including "quantum corrections").

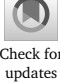
# 1   Introduction

The dynamics of integrable quantum many-body systems prepared in inhomogeneous states have attracted an increasing attention for more than two decades [1–22], arguably for their potentiality to elucidate basic questions of quantum transport. The last years in particular have seen a lot of progress in the physical and technical understanding of such dynamics. On the one hand, universal properties have been uncovered within conformal field theories or, more generally, studying the time evolution of states sufficiently close to ground states of critical systems [23–34]; on the other hand, a generalised hydrodynamic description (GHD) has been shown to capture the behaviour of integrable systems at large scales [35,36]. GHD was originally based on a conjecture on the expectation value of the currents in stationary states and was tested by studying both entanglement entropies [37–44] and correlation functions [45–65, 67–74]. Remarkably, the theory has been also verified in one experiment [75]. More recently, analytical proofs of its validity in specific integrable models have appeared [76,77].

First-order GHD looks like a collection of standard continuity equations for particle densities; the big step that delayed its discovery for several years was however the identification of the velocities entering the continuity equations. The main complication that arises in the presence of interactions is indeed the impossibility to define particle velocities independently of the state of the system [78]: the triviality of GHD is only apparent, as the velocities satisfy other equations coupled in a nonlinear way to the GHD ones.

Since its discovery, it has been evident that first-order GHD is a very powerful tool. It has however some limitations, and it was often blamed to be valid only in limits in which the system behaves essentially in a classical way (indeed, $\hbar$ does not appear in first-order GHD). In addition, the theory is unable to capture diffusive behaviour in interacting systems, a subject that has been attracting more and more attention [66,67,70,71,79–81]. In order to overcame this weakness, there have been proposals to go beyond the infinite time/low inhomogeneity limit of the original formulation in interacting integrable systems [79, 80, 82, 83], but, at the same time, some issues in noninteracting spin chains were uncovered. Specifically, the attempt of Ref. [84] to develop a higher-order GHD in noninteracting spin chains was unsuccessful, notwithstanding the author made use of the exact solution to the dynamics (which, inciden-

tally, will be proven in this paper). We will show that such attempt has been spoiled by an inappropriate definition of space dependent root densities, a concept that was introduced in GHD without a definition that could have been directly lifted into a nonperturbative level. The problem is that a generic homogeneous quantum state can not be characterised solely by the so-called root densities [86], and, in principle, one has also to define additional fields that take care of the off-diagonal matrix elements of the density matrix. In the inhomogeneous setting, the distinction between root densities and auxiliary fields is not transparent, and, without a proper definition, such fields couple to the root densities in an obscure way. This issue is evident in interacting integrable systems, as the aforementioned additional fields do not even appear in the thermodynamic Bethe Ansatz [86,87], which is arguably the framework of GHD. Despite not necessarily hindering the study of second order GHD, as Refs [79,80,82,83] testify, the lack of an explicit connection with the actual quantum state makes it almost impossible to exhibit an inhomogeneous initial state that could be described by such equations: how can one be sure that the additional fields vanish in the initial state and that their contribution die out in a shorter time scale?

The main goal of this work is to show that some of the problems in going beyond first-order GHD can be traced back to the presence of ambiguities in the definitions of the quantities used to describe the dynamics. We propose a resolution of them in noninteracting spin chains, and this will allow us to make the framework of GHD precise and exact. We will comment in Section 6 on the interacting case.

## 1.1 Ambiguities in GHD

Generalised hydrodynamics was introduced as a large-scale description of time evolution in one dimensional integrable systems with inhomogeneities. Roughly speaking, at large times only the slowest degrees of freedom keep varying and the state becomes locally equivalent to a stationary state. Such a quasi-stationary state was called LQSS (locally quasi-stationary state) [69], and it was shown to be well described by the continuity equations satisfied by the charge densities in the limit of low inhomogeneity. GHD was simultaneously developed in quantum field theories [35] and quantum spin chains [36], but we restrict here to spin chains.

The basic elements of GHD are the additive charges $Q = \mathfrak{a} \sum_\ell Q_\ell$, commuting with the Hamiltonian $H = \mathfrak{a} \sum_\ell H_\ell$, and their corresponding currents $J[Q_j] = \mathfrak{a} \sum_\ell J_\ell[Q_j]$. Here $\mathfrak{a}$ denotes the lattice spacing (in the next section, its definition will be slightly modified) and the operators $O_\ell$ are quasi-localised around site $\ell$[1]. In the Heisenberg picture, charges and currents are connected by a continuity equation of the form

$$\partial_t Q_\ell + \frac{J_\ell[Q_j] - J_{\ell-1}[Q_j]}{\mathfrak{a}} = 0 \,. \tag{1}$$

Here we made a (asymmetric) choice for the position of the currents; we will reconsider the possibility of other conventions later. It is clear that we are free to add any constant in the definition of $J_\ell[Q_j]$; we choose the convention of imposing null trace, $\mathrm{tr}[J_\ell[Q_j]] = 0$, which corresponds to the vanishing of the currents at infinite temperature. These equations are sufficient to determine the current for given charge density, but the definition of the latter is still ambiguous. Indeed, the charge density can be always redefined as follows

$$Q_\ell^{[G]} = Q_\ell - G_\ell + G_{\ell-1} \,, \tag{2}$$

---

[1]A spin operator $O_\ell$ is quasi-localised around a given site $\ell$ if there is a sequence of connected subsystems $S_n \subset S_{n+1}$ ($S_n$ is a proper subset of $S_{n+1}$), centred around $\ell$, such that $\left\| O_\ell - \frac{\mathrm{tr}_{\overline{S_n}}[O_\ell]}{\mathrm{tr}_{\overline{S_n}}[\mathbb{I}]} \otimes \mathbb{I}_{\overline{S_n}} \right\| < e^{-\alpha |S_n|}$, with $\alpha > 0$ and $|S_n|$ the extent of $S_n$. A quasilocal charge is an additive charge with quasi-localised density - see Ref. [85] for a review on quasilocal charges in integrable spin chains.

where $G_\ell$ is a quasi-localised operator acting around site $\ell$. This redefinition produces a change not only in the current

$$J_\ell[Q_j^{[G]}] = J_\ell[Q_j] + i\mathfrak{a}[H, G_\ell], \tag{3}$$

but also in the integrated current

$$J[Q_j^{[G]}] = J[Q_j] + i\mathfrak{a}[H, G]. \tag{4}$$

Because of this, the choice of $G = \mathfrak{a} \sum_\ell G_\ell$ and of its density $G_\ell$ is not irrelevant, especially if one is interested in going beyond first-order GHD. Depending on $G$, indeed, the resulting current can be conserved or not. In the former case, if the expectation values of the charges can be expressed in terms of densities, somehow describing the diagonal part of the density matrix, the expectation values of the currents would be as well. Non-conserved currents would have instead also off-diagonal matrix elements, which, through the continuity equation (1), would generate also off-diagonal contributions in the expectation values of the charges in inhomogeneous states.

Assuming that one can find a conserved operator $C[Q_j]$ such that the following operator is quasilocal,

$$C[Q_j] + i\mathfrak{a}^{-1} \sum_{E \neq E'} \frac{\langle E|J[Q_j]|E'\rangle}{E - E'} |E\rangle \langle E'| \qquad (H|E\rangle = E|E\rangle), \tag{5}$$

we propose to set $G$ equal to (5) so that the integrated current becomes conserved

$$\mathfrak{a}[H, [H, G]] = i[H, J[Q_j]]. \tag{6}$$

We denote it by $J[Q]$ to emphasise that it is now just a functional of the charge and not of its particular density: whatever the local charge density $Q_j$ is defined, we have

$$J[Q] = \sum_E \langle E|J[Q_j]|E\rangle |E\rangle \langle E|. \tag{7}$$

Note that the current is still defined up to the density of a charge $C[Q_j]$, which will be chosen in such a way to make the current of the current conserved, and so on and so forth[2]. This procedure defines an invariant subspace of operators, linear combinations of charge densities, that we call *locally quasi-conserved operators*, LQCO. The density matrix of an LQSS can then be defined as the exponential of an LQCO.

Generally, in noninteracting spin chains with local Hamiltonians, the standard choice for the local conservation laws [45, 88, 89] is such that (5) is quasilocal even for $C[Q_j] = 0$; thus, it is reasonable to expect that (6) could be enforced. We will see that (6) plays a key role in the inhomogeneous generalisation of the so-called *root densities*, which were originally introduced to characterise stationary states in integrable models that allow for a thermodynamic Bethe Ansatz description [86, 87]. Specifically, it allows us to define root densities that depend on the site and that completely determine the expectation values of the charge densities and vice versa.

We mention that similar issues have been already discussed [77, 80]. In particular, in the derivation of a diffusive correction to GHD, refs [79, 80] have proposed to choose a $\mathcal{PT}$-invariant gauge, which turns out to be unambiguous once requiring the expectation values of the charge densities to be real in stationary states. Our gauge choice, on the other hand, is not driven by uniqueness: we aim at exploiting the degrees of freedom in the definitions of

---

[2]The remaining gauge invariance encoded in $C[Q_j]$ can be used to make the expectation values of the charge densities as closer as possible in form to their expectation values in the homogeneous case

the charge densities in such a way to decouple the dynamical equations describing the time evolution of the locally quasi-stationary states from the rest.

Finally, we stress that the ambiguities pointed out in this section must be lifted in order to compare analytical predictions against numerical (or experimental) data, where following different conventions could result in artificial discrepancies. Besides, unfortunate conventions could introduce cumbersome large-time corrections that could be otherwise reabsorbed in the definitions of the operators. This is becuase, as we will see, the auxiliary fields characterising the state together with the root densities satisfy dynamical equations different from the GHD ones, so their presence in the initial state affects the large time corrections. For example, Ref. [84] could not capture a universal part of the light-cone dynamics within the proposed higher-order GHD. The problem was not in the universality of the phenomenon, while in having defined space-dependent root densities in an inappropriate way. Specifically, the inhomogeneous generalisation of the root density proposed in Ref. [84] is the quantity that, in the next section, will be called "spuriously local root density". While the latter has the correct homogeneous limit, in the presence of inhomogeneities it is not dynamically decoupled from the remaining degrees of freedom. To overcome this problem and write a dynamical equation written solely in terms of the root density, Ref. [84] projected at every time on the subspace of states that were completely characterised by the spuriously local root density. The resulting equation was not supposed to be exact, it was instead questioned whether it could nevertheless be valid at the next order in the inhomogeneity. The answer was disappointedly negative. In this paper we will not make any approximation and show how to fix this issue.

## 1.2 Outline

Section 2 - *Preliminaries* - introduces the systems under investigation and a convenient representation of states and operators.

Section 3 - *Time evolution of inhomogeneous states* - collects the main results. In particular, an invariant subspace made of LQSSs is identified, and the equations of motion governing time evolution within that subspace are exhibited (for homogeneous Hamiltonians, they are also solved) and identified with generalised hydrodynamics. We show that the complementary subspace is invariant as well and exhibit the corresponding dynamical equations.

Section 4 - *The two-temperature scenario revisited* - details the solution to the dynamics in one of the most studied settings. The predictions are checked against numerical data.

Section 5 - *Continuum scaling limit* - compares the lattice dynamics with the dynamics in the quantum field theory emerging in the limit where the lattice spacing $\mathfrak{a}$ approaches 0.

Section 6 - *Conclusion* - is a collection of observations.

Appendix A - *Non-equilibrium dynamics in inhomogeneous systems* - has a proof of the Moyal dynamical equation in noninteracting spin chains.

Appendix B - *Weak inhomogeneous limit: an asymptotic expansion* - presents a perturbative derivation of the equations of motion through a formal asymptotic expansion of the exact solution in the limit where a parameter characterising the inhomogeneity approaches infinity (homogeneous limit).

## 1.3 List of the notations

$[A, B]_{\pm} = AB \pm BA$ denote the anticommutator and the commutator, respectively.

$\mathbb{Z}$, $\mathbb{N}_0$, $\frac{1}{2}\mathbb{Z}$, and $\frac{1}{2} + \mathbb{Z}$ are the set of the integers, the set of the nonnegative integers, the set of the integers and the half-integers, and the set of the half-integers, respectively.

$\boldsymbol{O}$, in a bold capital letter, is a linear operator acting on the spin chain.

$\mathcal{O}$, in a calligraphic capital letter, is the matrix associated with the noninteracting operator that is indicated with the same capital letter in bold (*cf.* (13)), *i.e.*, $\boldsymbol{O}$.

$\hat{o}$, with a hat on a lower case letter, denotes the symbol of the matrix indicated with the same letter, in a calligraphic upper case (*cf.* (13) and (14)), *i.e.*, $\mathcal{O}$. We also call it the symbol of the operator indicated with the corresponding bold capital letter, *i.e.*, $\boldsymbol{O}$.

$\hat{o}^{\text{phys}}$ is the physical part of $\hat{o}$, which completely characterizes $\boldsymbol{O}$.

$\hat{o}^{\pm}(e^{ip})$ are the $\pi$-periodic and the $\pi$-anti-periodic parts of $\hat{o}$ with respect to $\mathfrak{a}p$.

$\hat{o}^{(x_0)}$ is the symbol of an operator $\boldsymbol{O}^{(x_0)}$ localized around $x_0$.

$2\kappa$ is the dimension of the representation of the symbol.

$\mathfrak{a}$ denotes $\kappa$ times the lattice spacing.

$\star$ denotes the Moyal star product.

$x$ (sometimes $y$) and $t$ are the position and the time. They generally appear as subscripts. If the quantity of interest does not depend on $x$ and/or $t$, the letter is simply not shown.

$p$ (sometimes $q$) is the momentum. In the symbols, it appears in the form of a phase $z_p = e^{i\mathfrak{a}p}$ to emphasise that the symbol is a $2\pi$-periodic function of $\mathfrak{a}p$.

$\varepsilon$, $v$, $\theta$, and $\phi$ are the dispersion relation, the velocity of the elementary excitations, the Bogoliubov angle, and the angle between the axis of the Bogoliubov rotation and the $z$ axis (in the quantum XY model $\phi = 0$), respectively, for $\kappa = 1$.

$\Gamma$ is the correlation matrix and $\hat{\Gamma}$ is its symbol (this is the only exception to the convention for which symbols are represented by lower-case letters).

$\rho$ is the root density for $\kappa = 1$.

e/o in a subscript selects the even/odd part of the quantity with respect to the momentum.

R/I in a subscript selects the real/imaginary part of the quantity.

## 2 Preliminaries

### 2.1 The Hamiltonian

We consider the nonequilibrium time evolution generated by a noninteracting spin-chain Hamiltonian whose density is almost invariant under a shift by $\kappa$ sites. This class of Hamiltonians includes:

a) the quantum Ising model [90, 91] ($\kappa = 1$)

$$H_{\text{Ising}} = -J \sum_{\ell \in \mathbb{Z}} \sigma_\ell^x \sigma_{\ell+1}^x + g \sigma_\ell^z, \tag{8}$$

where $\sigma_\ell^\alpha$ act like Pauli matrices on site $\ell$ and like the identity elsewhere, and $\mathbb{Z}$ is the set of all the integers;

b) the XX model [91,92] ($\kappa = 1$)

$$H_{\text{XX}} = J \sum_{\ell \in \mathbb{Z}} \sigma_\ell^x \sigma_{\ell+1}^x + \sigma_\ell^y \sigma_{\ell+1}^y + g\sigma_\ell^z ; \tag{9}$$

c) the (noninteracting) spin-Peierls model [93] ($\kappa = 2$)

$$H_{\text{Peierls}} = J \sum_{\ell \in \mathbb{Z}} (1 + \delta(-1)^\ell)(\sigma_\ell^x \sigma_{\ell+1}^x + \sigma_\ell^y \sigma_{\ell+1}^y) + \alpha(\sigma_\ell^x \sigma_{\ell+1}^z \sigma_{\ell+2}^x + \sigma_\ell^y \sigma_{\ell+1}^z \sigma_{\ell+2}^y). \tag{10}$$

In its most general form, the Hamiltonian is as follows (the homogeneous case was introduced in Ref. [94])

$$H = \sum_{\ell \in \mathbb{Z}} \sum_{i,j=1}^\kappa \sum_{\alpha,\beta \in \{x,y\}} \sum_{n \in \mathbb{N}_0} J_{n;i,j}^{\ell;\alpha\beta} \sigma_{\kappa\ell+i}^\alpha \prod_{m=\kappa\ell+i+1}^{\kappa(\ell+n)+j-1} \sigma_m^z \sigma_{\kappa(\ell+n)+j}^\beta + \sum_{\ell \in \mathbb{Z}} \sum_{i=1}^\kappa J_i^{\ell;z} \sigma_{\kappa\ell+i}^z , \tag{11}$$

where $\mathbb{N}_0$ is the set of the nonnegative integers. The coupling constants can be arbitrary, but, in almost the entire paper, we will assume that $J_{n;i,j}^{\ell;\alpha\beta}$ and $J_i^{\ell;z}$ do not depend on $\ell$.

Under the Jordan-Wigner transformation

$$a_{2\ell-1} = \prod_{j<\ell} \sigma_j^z \sigma_\ell^x \qquad a_{2\ell} = \prod_{j<\ell} \sigma_j^z \sigma_\ell^y , \qquad [a_\ell, a_n]_+ = 2\delta_{\ell n} \mathbf{I}, \tag{12}$$

where $[A,B]_+ = AB + BA$ denotes the anticommutator, the Hamiltonian is mapped into a chain of noninteracting Majorana fermions, as follows

$$H = \frac{1}{4} \sum_{\ell,n \in \mathbb{Z}} \sum_{i,j=1}^{2\kappa} \mathcal{H}_{ij}^{\ell n} a_{2\kappa\ell+i} a_{2\kappa n+j} . \tag{13}$$

Here $\mathcal{H}$ is an infinite purely imaginary Hermitian matrix $(\mathcal{H}_{n\ell}^{ji})^* = \mathcal{H}_{\ell n}^{ij} = -\mathcal{H}_{n\ell}^{ji}$. We point out that, contrary to standard conventions, we call $\mathcal{H}$ a matrix despite its indices not being strictly positive.

If the coupling constants are smooth functions of the position, the Hamiltonian is approximately translationally invariant in any sufficiently small region; this property can be exploited by writing $\mathcal{H}$ in a "local" Fourier space

$$H = \frac{1}{4} \sum_{r \in \frac{1}{2}\mathbb{Z}} \sum_{n \in \mathbb{Z}} \sum_{i,j=1}^{2\kappa} \int_{-\pi}^{\pi} \frac{\mathrm{d}[p\mathfrak{a}/\hbar]}{2\pi} e^{2i\frac{(r-n)\mathfrak{a}p}{\hbar}} \big[\hat{h}_{r\mathfrak{a}}(e^{i\mathfrak{a}p/\hbar})\big]_{ij} a_{2\kappa(2r-n)+i} a_{2\kappa n+j} . \tag{14}$$

Here $\frac{1}{2}\mathbb{Z}$ is the set consisting of the integers and the half-integers; $\hbar$ is the reduced Planck constant and $\mathfrak{a}$ is $\kappa$ times the lattice spacing. We point out that $\kappa$ can be any positive integer, but, in the (quasi) homogeneous case, it is conveniently chosen to be proportional to the number of sites the system is (almost) invariant under a shift of.

**The symbol.** The ($2\kappa$)-by-($2\kappa$) matrix $\hat{h}_x(e^{i\mathfrak{a}p/\hbar})$ represents the Hamiltonian in the Fourier space around position $x$; we call it 'symbol', by analogy with the definition of the symbol of (block-)Toeplitz and (block-)circulant matrices. It will be clearer later that $p$ plays the role of the conjugate variable of $x$, so we call it momentum.

We note that, for given $x$, half of the Fourier components of $\hat{h}_x(e^{i\mathfrak{a}p/\hbar})$ do not enter the expression. In particular, for (half-)integer $x/\mathfrak{a}$, the relevant part of the symbol, which we call $\hat{h}_x^{\text{phys}}(e^{i\mathfrak{a}p/\hbar})$, is a $\pi$-(anti-)periodic matrix function of $\mathfrak{a}p/\hbar$. The remaining Fourier components

can be chosen arbitrarily, and hence they will be referred to as "unphysical". Nevertheless, we require the symbol to remain Hermitian and to satisfy $[\hat{h}_x(e^{i\mathfrak{a}p/\hbar})]^t = -\hat{h}_x(e^{-i\mathfrak{a}p/\hbar})$, where $^t$ denotes transposition. We also define $\hat{h}_x^{\pm}(e^{i\mathfrak{a}p/\hbar})$ as the extensions of the symbols obtained interpolating either the functions evaluated at $\frac{x}{\mathfrak{a}} \in \mathbb{Z}$ or at $\frac{x}{\mathfrak{a}} \in \frac{1}{2} + \mathbb{Z}$, the latter being the set of the half-integers. Thus we have

$$
\begin{aligned}
\hat{h}_x^{\pm}(-e^{i\mathfrak{a}p/\hbar}) &= \pm\hat{h}_x^{\pm}(e^{i\mathfrak{a}p/\hbar}) \\
\hat{h}_x^{\mathrm{phys}}(e^{i\mathfrak{a}p/\hbar}) &= \hat{h}_x^{(-1)^{2x}}(e^{i\mathfrak{a}p/\hbar}) \qquad \frac{x}{\mathfrak{a}} \in \frac{1}{2}\mathbb{Z}.
\end{aligned}
\tag{15}
$$

A possible extension to $\frac{x}{\mathfrak{a}} \in \mathbb{R}$ is the following

$$
\begin{aligned}
\hat{h}_x(e^{i\mathfrak{a}p/\hbar}) &= \sum_{\frac{y}{\mathfrak{a}} \in \frac{1}{2}\mathbb{Z}} \frac{\sin\left(\pi\frac{x-y}{\mathfrak{a}}\right)}{\pi\frac{x-y}{\mathfrak{a}}} \hat{h}_y^{\mathrm{phys}}(e^{i\mathfrak{a}p/\hbar}) \\
\hat{h}_x^{\pm}(e^{i\mathfrak{a}p/\hbar}) &= \sum_{\frac{y}{\mathfrak{a}} \in \frac{1}{2}\mathbb{Z}} \frac{1 \pm (-1)^{\frac{2y}{\mathfrak{a}}}}{2} \frac{\sin\left(\pi\frac{x-y}{\mathfrak{a}}\right)}{\pi\frac{x-y}{\mathfrak{a}}} \hat{h}_y^{\mathrm{phys}}(e^{i\mathfrak{a}p/\hbar}).
\end{aligned}
\tag{16}
$$

Using these definitions, $\hat{h}_x(e^{i\mathfrak{a}p/\hbar})$ and $\hat{h}_x^{(\pm)}(e^{i\mathfrak{a}p/\hbar})$ are *entire* (matrix-)functions of $x$.

The formal expressions that will be derived in this paper (and that can be applied to any noninteracting spin chain) will be explicitly worked out in a generalised XY model with $\kappa = 1$. In the homogeneous case, such model describes a *generic one-site shift invariant noninteracting spin-chain Hamiltonian*. Its symbol will be parametrised as follows:

$$
\hat{h}(e^{i\mathfrak{a}p/\hbar}) = e^{-i\frac{\phi(p)}{2}\sigma^y} e^{-i\frac{\theta(p)}{2}\sigma^z} [\varepsilon_{\mathrm{o}}(p)\mathrm{I} + \varepsilon_{\mathrm{e}}(p)\sigma^y] e^{i\frac{\theta(p)}{2}\sigma^z} e^{i\frac{\phi(p)}{2}\sigma^y},
\tag{17}
$$

where the subscripts 'e' and 'o' stand for the even and the odd part respectively (with respect to the momentum $p$); $\varepsilon(p)$ is the dispersion relation; $\theta(p) = -\theta(-p)$ is the Bogoliubov angle and $\phi(p) = \phi(-p)$ parametrises the angle of the axis of the Bogoliubov rotation. When we want to emphasize the dependence on the phase, we write $\theta(p) = \Theta(e^{i\mathfrak{a}p/\hbar})$. For example, in the quantum Ising model (a) we have $\varepsilon(p) = 2J\sqrt{1 + g^2 - 2g\cos(\mathfrak{a}p/\hbar)}$, $e^{i\theta(p)} = \frac{2J(e^{i\mathfrak{a}p/\hbar} - g)}{\varepsilon(p)}$ (*i.e.*, $e^{i\Theta(z)} = z^{\frac{1}{2}}\frac{\sqrt{z-g}}{\sqrt{1-gz}}$), and $\phi(p) = 0$. If not stated otherwise, all the quantities will be assumed to be smooth functions of the momentum, which generally happens in (quasi)local noncritical Hamiltonians.

A convenient parametrization for inhomogeneous Hamiltonians will be shown later - (72).

## 2.2 The state

The systems considered in this paper are prepared in a gaussian state, *i.e.*, a state where the Wick's theorem can be applied to the correlations of the Majorana fermions (12)

$$
\langle \boldsymbol{a}_{j_1} \boldsymbol{a}_{j_2} \cdots \boldsymbol{a}_{j_n} \rangle = \mathrm{pf}[A], \qquad A_{\ell n} = \langle \boldsymbol{a}_{j_\ell} \boldsymbol{a}_{j_n} \rangle - \delta_{j_\ell j_n},
\tag{18}
$$

'pf' denoting the Pfaffian. This can be the ground state (if a symmetry is not spontaneously broken) or an excited state of a noninteracting spin-chain Hamiltonian, like (11), as well as a thermal state or a generalised Gibbs ensemble [95–97]. Since the dynamics are noninteracting, Wick's theorem holds at any time, and the expectation values of the observables can be written in terms of the correlation matrix

$$
\Gamma_{ij}^{\ell n}(t) = \delta_{\ell n}\delta_{ij} - \langle a_{2\kappa\ell+i} a_{2\kappa n+j} \rangle_t.
\tag{19}
$$

Here $\langle\cdot\rangle_t$ denotes the expectation value at time $t$. As done for the quadratic operators, we define the physical part of the symbol $\hat{\Gamma}_{x,t}(e^{i\mathfrak{a}p})$ of the correlation matrix around position $x$ as follows

$$[\hat{\Gamma}^{\text{phys}}_{x,t}(z_p)]_{ij} = \sum_{\ell\in\mathbb{Z}} e^{-2i\frac{(\ell\mathfrak{a}-x)p}{\hbar}}\Gamma^{\ell,\frac{2x}{\mathfrak{a}}-\ell}_{ij}(t) \qquad \frac{x}{\mathfrak{a}}\in\frac{1}{2}\mathbb{Z}, \tag{20}$$

where $z_p = e^{i\mathfrak{a}p/\hbar}$. At a given time, we can extend (20) so as to allow for real $x$ using, for example, (16); however, we will not impose (16) at a generic time, being more convenient to exploit such degrees of freedom to simplify the equations of motion - Appendix A. Nevertheless, we will assume that the extension remains an entire function of the position at any time.

For $\kappa = 1$ (which is a convenient representation when the state is almost invariant under a shift by one site) the state is completely characterised by a spuriously local root density $\rho^{\text{fake}}_{x,t}(p)$, and by a spuriously local auxiliary complex field $\Psi^{\text{fake}}_{x,t}(p) = -\Psi^{\text{fake}}_{x,t}(-p)$ as follows

$$\hat{\Gamma}_{x,t}(z_p) = 4\pi\hbar e^{-i\frac{\phi(p)}{2}\sigma^y}e^{-i\frac{\theta(p)}{2}\sigma^z}\bigg[\rho^{\text{fake}}_{x,t;\text{o}}(p)\text{I} + \bigg(\rho^{\text{fake}}_{x,t;\text{e}}(p) - \frac{1}{4\pi\hbar}\bigg)\sigma^y +$$
$$\Psi^{\text{fake}}_{x,t;\text{R}}(p)\sigma_z - \Psi^{\text{fake}}_{x,t;\text{I}}(p)\sigma^x\bigg]e^{i\frac{\theta(p)}{2}\sigma^z}e^{i\frac{\phi(p)}{2}\sigma^y}. \tag{21}$$

This representation, which at first glance could look bizarre (for the appearance of the angles $\theta(p)$ and $\phi(p)$, which are properties of the Hamiltonian and not of the state), allows us in fact to attribute a physical meaning to $\rho^{\text{fake}}_{x,t}(p)$ and $\Psi^{\text{fake}}_{x,t}(p)$: in the homogeneous limit, $\rho^{\text{fake}}(p)$ becomes the density in phase space $\delta x\,\mathrm{d}p$ of the excitations over the ground state of the quasi-particle excitations (provided that $\varepsilon(p) \geq 0$) [45], and $\Psi^{\text{fake}}(p)$ describes the off-diagonal part of the density matrix. On the other hand, such properties are lost in the presence of inhomogeneities. Indeed Ref. [84] showed that a higher-order hydrodynamics based on such spurious quantities can not be exact. The main aim of this paper is to construct a genuinely local version of these quantities, which we will simply call $\rho(p)$ and $\Psi(p)$. They will be chosen so as to accomplish the goal outlined in 1.1; in particular, they will reduce to their global counterparts in the homogeneous limit, and they will transform as simply as possible under time evolution.

We note that, in the homogeneous limit, the root density and the auxiliary field satisfy the following constraints (the maximal eigenvalue of $\hat{\Gamma}(e^{i\mathfrak{a}p})$ in absolute value can not exceed 1)

$$\begin{aligned}0 \leq \rho(p) &\leq \frac{1}{2\pi\hbar}\\ |\Psi(p)|^2 &\leq \min(\rho(p),\rho(-p))\bigg[\frac{1}{2\pi\hbar}-\max(\rho(p),\rho(-p))\bigg].\end{aligned} \tag{22}$$

These bounds are not always satisfied in inhomogeneous systems.

## 2.3 Expectation values and local operators

The expectation value of a traceless quadratic operator $O$ can be expressed in terms of its symbol and of the symbol of the correlation matrix as follows

$$\langle O\rangle_t = \frac{\text{tr}[\Gamma(t)\mathcal{O}]}{4} = \sum_{\frac{x}{\mathfrak{a}}\in\frac{1}{2}\mathbb{Z}}\int_{-\pi}^{\pi}\frac{\mathrm{d}[\mathfrak{a}p/\hbar]}{8\pi\kappa}\text{tr}[\hat{\Gamma}^{\text{phys}}_{x,t}(z_p)\hat{o}_x(z_p)] \equiv$$
$$\sum_{\frac{x}{\mathfrak{a}}\in\frac{1}{2}\mathbb{Z}}\int_{-\pi}^{\pi}\frac{\mathrm{d}[\mathfrak{a}p/\hbar]}{8\pi\kappa}\text{tr}[\hat{\Gamma}_{x,t}(z_p)\hat{o}^{\text{phys}}_x(z_p)]. \tag{23}$$

These equations are exact; the drawback is that they involve the physical part of one of the two symbols; in Section 3.3 we will exhibit an explicitly symmetrical expression. We can express the symbol of operators (quasi)localised around $\ell$ as follows:

$$\hat{o}_x^{(\ell)\,\mathrm{phys}}(z_p) = \hat{o}_{e_1}(z_p, e^{\partial_\ell})\delta_{x,\ell\mathfrak{a}} + \hat{o}_{o_1}(z_p, e^{\partial_\ell})\delta_{x,(\ell+\frac{1}{2})\mathfrak{a}}, \tag{24}$$

where $e_1$ and $o_1$ select the even and the odd part with respect to the first argument. The symbol of the corresponding translationally invariant operator is

$$\hat{o}(z_p) = \hat{o}(z_p, 1). \tag{25}$$

For $\kappa = 1$ the auxiliary function will be parametrised as follows

$$\hat{o}(z_p, w) = e^{-i\frac{\phi(p)}{2}\sigma^y}e^{-i\frac{\theta(p)}{2}\sigma^z}\left[\omega(p,w)\frac{I+\sigma^y}{2} - \omega(-p,w)\frac{I-\sigma^y}{2} + \left(\Omega_{o_1}(p,w)\frac{\sigma^z-i\sigma^x}{2} + h.c.\right)\right]e^{i\frac{\theta(p)}{2}\sigma^z}e^{i\frac{\phi(p)}{2}\sigma^y}. \tag{26}$$

In terms of the spuriously local root density and spuriously local auxiliary field the expectation value reads as

$$\langle \boldsymbol{O}_\ell | \boldsymbol{O}_\ell \rangle_t = \sum_{s=\pm 1}\int_{-\pi}^{\pi} d[\mathfrak{a}p]\,\omega^{(s)}(p, e^{\partial_\ell})\left(\rho_{\ell+\frac{1-s}{2}}^{\mathrm{fake}}(p) - \frac{1}{4\pi}\right) + \left(\Omega_{o_1}^{(s)}(p, e^{\partial_\ell})\Psi_{\ell+\frac{1-s}{2}}^{\mathrm{fake}}(p) + h.c.\right). \tag{27}$$

One of the goals of this paper is to give a convenient definition of charge density so that its expectation value has a form similar to (27) with the spuriously local root density and auxiliary field replaced by the analogous genuinely local quantities.

## 2.4 Reduced density matrix

The reduced density matrix (RDM) of a connected subsystem $A \equiv [x_1, x_2]$ in a gaussian state is gaussian. Its correlation matrix is obtained by restricting the correlation matrix of the state to the subsystem. Within the Wigner description that we introduced, it is convenient to embed the RDM into the entire system as follows:

$$\mathrm{tr}_{\overline{A}}[\boldsymbol{\rho}] \to \boldsymbol{\rho}_A^{ext} \equiv \mathrm{tr}_{\overline{A}}[\boldsymbol{\rho}] \otimes \frac{I_{\overline{A}}}{2^{|\overline{A}|}}, \tag{28}$$

which has a correlation matrix that zeros outside $A$. The symbol of $\boldsymbol{\rho}_A^{ext}$ is then given by

$$\hat{\Gamma}_{x,t}^{[x_1,x_2]}(z_p) = \theta_H(x_1 \le x \le x_2)\int_{-\pi}^{\pi}\frac{d[\mathfrak{a}\wp/\hbar]}{2\pi}\frac{\sin\left(\left(\frac{x_2-x_1-|2x-x_1-x_2|}{\mathfrak{a}} + \frac{1}{2}\right)\mathfrak{a}\wp/\hbar\right)}{\sin\left(\frac{1}{2}\mathfrak{a}\wp/\hbar\right)}\hat{\Gamma}_{x,t}(z_{p-\wp}). \tag{29}$$

For $x_1 \ll x \ll x_2$, the distribution in convolution with the symbol approaches a Dirac delta function, so, as expected, the bulk is not affected by the boundaries of the subsystem.

We refer the reader to Ref. [45] for more examples and for an overview of the description in terms of symbols in homogeneous noninteracting spin chains.

# 3  Time evolution of inhomogeneous states

It is well known that the dynamics generated by quadratic operators have infinitely many invariant subspaces, indeed it conserves the "number" of the Majorana fermions[3]

$$e^{iHt}\boldsymbol{a}_\ell e^{-iHt} = \sum_{n\in\mathbb{Z}} [e^{-i\mathcal{H}t}]_{\ell n}\boldsymbol{a}_n. \tag{30}$$

In this section we show that, in fact, it is reducible even more, and one of the invariant subspaces consists of the locally quasi-stationary states, qualitatively introduced in Ref. [69].

In the following the reduced Planck constant $\hbar$ and the lattice spacing $\mathfrak{a}$ will be written explicitly so that the reader can identify quantum and lattice contributions more easily.

## 3.1  Moyal dynamical equation

The symbol of an inhomogeneous state time evolves according to the following Moyal dynamical equation

$$\boxed{i\hbar\partial_t\hat{\Gamma}_{x,t}(z_p) = \hat{h}_x(z_p)\star\hat{\Gamma}_{x,t}(z_p) - \hat{\Gamma}_{x,t}(z_p)\star\hat{h}_x(z_p),} \tag{31}$$

where $z_p = e^{i\mathfrak{a}p/\hbar}$, and $f_x(p)\star g_x(p)$ is the Moyal star product [98]

$$f_x(p)\star g_x(p) = e^{i\hbar\frac{\partial_q\partial_x - \partial_p\partial_y}{2}}f_x(p)g_y(q)\Big|_{\substack{q=p\\y=x}}. \tag{32}$$

A proof of (31) is reported in Appendix A. Using that the star product is associative, the solution to (31) can be written as

$$\hat{\Gamma}_{x,t}(z_p) = e_\star^{-it\hat{h}_x(z_p)/\hbar}\star\hat{\Gamma}_{x,0}(z_p)\star e_\star^{it\hat{h}_x(z_p)/\hbar}, \tag{33}$$

where

$$e_\star^{-it\hat{h}_x(z_p)/\hbar} = \sum_{n=0}^\infty \frac{(-it)^n}{\hbar^n n!}\underbrace{\hat{h}_x(z_p)\star\cdots\star\hat{h}_x(z_p)}_{n}. \tag{34}$$

This can be simplified if the Hamiltonian is homogeneous ($\hat{h}_x(z_p)$ is independent of $x$), as follows

$$\hat{\Gamma}_{x,t}(z_p) = e^{-\frac{i}{2}\hbar\partial_q\partial_x}e^{-it\hat{h}(z_{p+q})/\hbar}\hat{\Gamma}_{x,0}(z_p)e^{it\hat{h}(z_{p-q})/\hbar}\Big|_{q=0}. \tag{35}$$

The solution to (35) is readily obtained and reads as

$$\hat{\Gamma}_{x,t}(z_p) = \iint_{-\infty}^\infty \frac{\mathrm{d}y\,\mathrm{d}q}{2\pi\hbar}e^{i\frac{q(x-y)}{\hbar}}e^{-it\hat{h}(z_{p+q/2})/\hbar}\hat{\Gamma}_{y,0}(z_p)e^{it\hat{h}(z_{p-q/2})/\hbar}. \tag{36}$$

These equations are the fundamentals of a Wigner description [99] of the dynamics in noninteracting spin-chain models, and they generalise the well-known equations describing free particle systems, in the sense that functions are replaced by matrices of functions (a reader could be more comfortable with referring to the symbol of the correlation matrix as a 'block Wigner function'). They have been formerly announced in Ref. [84]. We stress that (31) and (36) apply to spin chains, despite the variable position $x$ not being discrete.

---

[3]Note that, in order to define a genuine number operator, it would be necessary to map the Majorana fermions into spinless fermions, for example, through the J-W mapping $c_\ell^\dagger = (a_{2\ell-1} + ia_{2\ell})/2$.

## 3.2 Invariant subspaces

The dynamics described by (31) are reducible. We identify two invariant subspaces: the inhomogeneous generalisation of stationary states, which, following the terminology of Ref. [69], will be called LQSSs, and a subspace of states that we call ODSs (off-diagonal states).

### 3.2.1 Locally quasi-stationary states

Let the Hamiltonian be translationally invariant. An LQSS has the following properties:

- it reduces to a stationary state in the homogeneous limit;

- it remains an LQSS under time evolution.

Imposing these conditions in the $\kappa = 1$ representation, we find that the state can be characterised by a Wigner function $\rho_x(p)$, which we identify with the localised version of the root density, and it appears in the symbol of the correlation matrix as follows

$$
\hat{\Gamma}_x^{\text{LQSS}}(z_p; G) =
$$
$$
4\pi\hbar\, e^{-i\frac{\phi\left(p-\frac{i\hbar\overrightarrow{\partial_x}}{2}\right)}{2}\sigma^y} e^{-i\frac{\theta\left(p-\frac{i\hbar\overrightarrow{\partial_x}}{2}\right)}{2}\sigma^z} \left\{ \rho_{x,o}^{(G)}(p) + \sigma^y\left[\rho_{x,e}^{(G)}(p) - \frac{1}{4\pi\hbar}\right] \right\} e^{i\frac{\theta\left(p+\frac{i\hbar\overleftarrow{\partial_x}}{2}\right)}{2}\sigma^z} e^{i\frac{\phi\left(p+\frac{i\hbar\overleftarrow{\partial_x}}{2}\right)}{2}\sigma^y},
$$
$$
\tag{37}
$$

with

$$
\rho_x(p) = G(z_p, e^{\frac{\mathfrak{a}\partial_y}{2}})\rho_{x+y}^{(G)}(p)\Big|_{y=0} \qquad G(z,1) = 1. \tag{38}
$$

In (37), the arrows on top of the derivatives indicate the direction where they act. Roughly speaking, $G$ parametrises a gauge invariance in the definition of the root density. For example, it includes the arbitrariness in associating a position to a quasi-localised operator. Because of $G$, the same root density can be used to describe different states, therefore, in order to avoid confusion, it must be fixed. If not stated otherwise, $G(z, w)$ will be set to unity and the notations will be eased, dropping the dependence on $G$ in the expressions, *e.g.* $\hat{\Gamma}_x^{\text{LQSS}}(z_p; 1) \to \hat{\Gamma}_x^{\text{LQSS}}(z_p)$. Crucially, contrary to standard conventions, the Wigner function that we have defined does not depend only on the state, but also on the Hamiltonian; this is one of the reasons why we prefer to call it 'root density', which is instead a quantity that is always defined for given Hamiltonian and reference state.

Note that only integer and half-integer positions appear in (37). In addition, for $G = 1$ the physical part of the root density is $\pi$ periodic at integer sites and $\pi$ anti-periodic at half-integer sites.

Finally, we remind the reader of a very well known property of Wigner functions: even if $\rho_x(p)$ is a physical root density at fixed $x$ (namely, $0 \leq \rho_x(p) \leq \frac{1}{2\pi\hbar}$), the LQSS associated with $\rho_x(p)$ could be unphysical, as its correlation matrix could have eigenvalues with modulus larger than 1. Thus, not every profile of charges is allowed in an LQSS.

### 3.2.2 Off-diagonal states

An ODS (off-diagonal state) has the following properties:

- it reduces to a state with vanishing diagonal elements in a basis of stationary states in the homogeneous limit;

- it remains an ODS under time evolution.

Imposing these conditions in the $\kappa = 1$ representation, we find that the state is completely characterised by an auxiliary field $\Psi_x(p)$, which appears in the symbol of the correlation matrix as follows

$$\hat{\Gamma}_x^{\text{ODS}}(z_p; G) =$$

$$4\pi\hbar\, e^{-i\frac{\phi\left(p-\frac{i\hbar\overrightarrow{\partial_x}}{2}\right)}{2}\sigma^y} e^{-i\frac{\theta\left(p-\frac{i\hbar\overrightarrow{\partial_x}}{2}\right)}{2}\sigma^z}\left\{\sigma^z\Psi_{x,\text{R}}^{(G)}(p) - \sigma^x\Psi_{x,\text{I}}^{(G)}(p)\right\} e^{i\frac{\theta\left(p+\frac{i\hbar\overleftarrow{\partial_x}}{2}\right)}{2}\sigma^z} e^{i\frac{\phi\left(p+\frac{i\hbar\overleftarrow{\partial_x}}{2}\right)}{2}\sigma^y}, \quad (39)$$

with

$$\Psi_x(p) = G\left(z_p, e^{\frac{\mathfrak{a}\partial_y}{2}}\right)\Psi_{x+y}^{(G)}(p)\Big|_{y=0} \qquad G(z, 1) = 1. \quad (40)$$

Also in this case, $G$ parametrises a gauge invariance in the definition of the local auxiliary field, with the same caveats pointed out in the LQSS case. If not stated otherwise, $G(z, w)$ will be set to unity.

### 3.2.3 Dynamics

Assuming a one-site shift invariant Hamiltonian, each state can be conveniently written in a $\kappa = 1$ representation, decomposing it in an LQSS part, characterised by a root density $\rho_x(p)$, plus an off-diagonal one, characterised by an auxiliary field $\Psi_x(p)$; the symbol of the correlation matrix can be written as follows:

$$\hat{\Gamma}_x(z_p; G, \tilde{G}) = 4\pi\hbar\, e^{-i\frac{\phi\left(p-\frac{i\hbar\overrightarrow{\partial_x}}{2}\right)}{2}\sigma^y} e^{-i\frac{\theta\left(p-\frac{i\hbar\overrightarrow{\partial_x}}{2}\right)}{2}\sigma^z} \times$$

$$\left\{I\rho_{x,\text{o}}^{(G)}(p) + \sigma^y\left[\rho_{x,\text{e}}^{(G)}(p) - \frac{1}{4\pi\hbar}\right] + \sigma^z\Psi_{x,\text{R}}^{(\tilde{G})}(p) - \sigma^x\Psi_{x,\text{I}}^{(\tilde{G})}(p)\right\} e^{i\frac{\theta\left(p+\frac{i\hbar\overleftarrow{\partial_x}}{2}\right)}{2}\sigma^z} e^{i\frac{\phi\left(p+\frac{i\hbar\overleftarrow{\partial_x}}{2}\right)}{2}\sigma^y}. \quad (41)$$

By definition, the root density degree of freedom remains decoupled from the auxiliary field one also at finite time; the dynamics are described by the decoupled Moyal equations

$$i\hbar\partial_t\rho_{x,t}(p) = \varepsilon(p) \star \rho_{x,t}(p) - \rho_{x,t}(p) \star \varepsilon(p) \quad (42)$$

$$i\hbar\partial_t\Psi_{x,t}(p) = -\varepsilon(p) \star \Psi_{x,t}(-p) + \Psi_{x,t}(p) \star \varepsilon(-p), \quad (43)$$

where, in the second line, we used the oddity of the auxiliary field to write the right hand side as an antisymmetric functional of $\varepsilon$ and $\Psi$. Equations (42) and (43) are arguably the main result of this paper for homogeneous Hamiltonians. In particular, they allow us to identify the (localised version of the) root density, as it appears in the symbol of the correlation matrix in (41), with a Wigner function even when there is a nontrivial Bogoliubov angle $\theta(p)$ around a direction characterised by $\phi(p)$. This means, for example, that they hold true also in the quantum Ising model. We postpone the proof of (42) and (43) to Section 3.4, where we consider the more general case of an inhomogeneous Hamiltonian. Equations (42) and (43) clearly show that the root density $\rho$ and the auxiliary field $\Psi$ characterise two separate subsets of states that are invariant under time evolution. Incidentally, we note that $\Psi_{x,t}(p) \star \Psi_{x,t}^*(p)$ and $\Psi_{x,t}^*(p) \star \Psi_{x,t}(p)$ satisfy the same equation of the root density.

The first orders of the asymptotic expansions of (42) and (43) in the limit of low inhomogeneity are

$$\partial_t\rho_{x,t}(p) = -v(p)\partial_x\rho_{x,t}(p) + \hbar^2\frac{v''(p)}{24}\partial_x^3\rho_{x,t}(p) - \hbar^4\frac{v^{\text{iv}}(p)}{1920}\partial_x^5\rho_{x,t}(p) + O(\hbar^6\partial_x^7)$$

$$i\hbar\partial_t\Psi_{x,t}(p) = 2\varepsilon_e(p)\Psi_{x,t}(p) - i\hbar v_e(p)\partial_x\Psi_{x,t}(p) - \hbar^2\frac{v_e'(p)}{4}\partial_x^2\Psi_{x,t}(p) \quad (44)$$

$$+ i\hbar^3\frac{v_e''(p)}{24}\partial_x^3\Psi_{x,t}(p) + O(\hbar^4\partial_x^4),$$

where $v(p) = \frac{d\varepsilon(p)}{dp}$ is the velocity of the quasiparticle excitation with momentum $p$. We note that the first equation truncated at $O(\partial_x)$ is the noninteracting version of first-order GHD [35, 36, 69]. It also corresponds to the classical limit $\hbar \to 0$, in which the root density satisfies a classical continuity equation. On the other hand, as long as the dispersion relation is not odd, $\Psi_{x,t}(p)$ describes purely quantum contributions. It is worth observing that the equations governing time evolution do not depend explicitly on the lattice spacing.

Equations (42) and (43) are solved by

$$\rho_{x,t}(p) = \iint_{-\infty}^{\infty} \frac{dy\,dq}{2\pi\hbar} e^{iq(x-y)/\hbar} e^{-it[\varepsilon(p+q/2)-\varepsilon(p-q/2)]/\hbar} \rho_{y,0}(p) \tag{45}$$

$$\Psi_{x,t}(p) = \iint_{-\infty}^{\infty} \frac{dy\,dq}{2\pi\hbar} e^{iq(x-y)/\hbar} e^{-it[\varepsilon(p+q/2)+\varepsilon(-p+q/2)]/\hbar} \Psi_{y,0}(p). \tag{46}$$

**Localisation.** The root density and the auxiliary field can be expressed in terms of the spurious quantities appearing in (21) by inverting (41). For the sake of simplicity, we do it explicitly only for the generalised XY model with $\phi(p) = 0$. We find

$$\rho_x^{(G)}(p) = C_-\left(z_p, e^{\frac{a\partial_x}{2}}\right)\rho_{x,o}^{fake}(p) + C_+\left(z_p, e^{\frac{a\partial_x}{2}}\right)\rho_{x,e}^{fake}(p) + \\ - iS_-\left(z_p, e^{\frac{a\partial_x}{2}}\right)\Psi_{x,R}^{fake}(p) - S_+\left(z_p, e^{\frac{a\partial_x}{2}}\right)\Psi_{x,I}^{fake}(p) \tag{47}$$

$$\Psi_x^{(\tilde{G})}(p) = C_-\left(z_p, e^{\frac{a\partial_x}{2}}\right)\Psi_{x,R}^{fake}(p) + iC_+\left(z_p, e^{\frac{a\partial_x}{2}}\right)\Psi_{x,I}^{fake}(p) + \\ - iS_-\left(z_p, e^{\frac{a\partial_x}{2}}\right)\rho_{x,o}^{fake}(p) + iS_+\left(z_p, e^{\frac{a\partial_x}{2}}\right)\rho_{x,e}^{fake}(p), \tag{48}$$

where

$$\begin{aligned} C_\pm(z,w) &= \cos\left(\frac{\Theta(z)-\Theta(zw)}{2} \pm \frac{\Theta(z)-\Theta(zw^{-1})}{2}\right) \\ S_\pm(z,w) &= \sin\left(\frac{\Theta(z)-\Theta(zw)}{2} \pm \frac{\Theta(z)-\Theta(zw^{-1})}{2}\right), \end{aligned} \tag{49}$$

and $\Theta(e^{iap/\hbar})$ is the Bogoliubov angle as defined below (17). We stress that the local root density at (half-)integer position is a mix of quantities defined at both integer and half-integer positions.

The first orders of the expansion in the limit of weak inhomogeneity read as

$$\rho_x^{(G)}(p) = \rho_x^{fake}(p) + \hbar\frac{\theta'(p)}{2}\partial_x\Psi_{x,R}^{fake}(p) + \hbar^2\frac{\theta'(p)^2}{8}\partial_x^2\rho_{x,o}^{fake}(p) - \hbar^2\frac{\theta''(p)}{8}\partial_x^2\Psi_{x,I}^{fake}(p) + \\ + O\left(\hbar^3\partial_x^3\right) \tag{50}$$

$$\Psi_x^{(\tilde{G})}(p) = \Psi_x^{fake}(p) + \hbar\frac{\theta'(p)}{2}\partial_x\rho_{x,o}^{fake}(p) + \hbar^2\frac{\theta'(p)^2}{8}\partial_x^2\Psi_{x,R}^{fake}(p) + i\hbar^2\frac{\theta''(p)}{8}\partial_x^2\rho_{x,e}^{fake}(p) + \\ + O\left(\hbar^3\partial_x^3\right).$$

In the XX model the Bogoliubov angle can be set to zero, therefore the spuriously local root densities are in fact genuinely local.

### 3.2.4 Locally quasi-conserved operators

If there are no special symmetries, in noninteracting spin chain models the conservation laws are quadratic forms of Majorana fermions, like the Hamiltonian in (14) [45]. In the Heisenberg picture, the matrices associated with quadratic operators time evolve like the correlation matrix, provided to reverse the sign of the time. This allows us to identify an invariant subspace of operators, which we call LQCOs (locally quasi-conserved operators), consisting of the quadratic forms of fermions with symbols of the form of an LQSS (37)

$$\hat{q}_x(z_p; G) = e^{-i\frac{\phi\left(p - \frac{i\hbar\overrightarrow{\partial_x}}{2}\right)}{2}\sigma^y} e^{-i\frac{\theta\left(p - \frac{i\hbar\overrightarrow{\partial_x}}{2}\right)}{2}\sigma^z} \left\{ q_{x,o}^{(G)}(p) + \sigma^y q_{x,e}^{(G)}(p) \right\} e^{i\frac{\theta\left(p + \frac{i\hbar\overleftarrow{\partial_x}}{2}\right)}{2}\sigma^z} e^{i\frac{\phi\left(p + \frac{i\hbar\overleftarrow{\partial_x}}{2}\right)}{2}\sigma^y}. \quad (51)$$

The analogue of the root density is the (single particle) eigenvalue's density in phase space, which can be defined as follows

$$q_x(p) = G(z_p, e^{\frac{\mathfrak{a}\partial_x}{2}}) q_x^{(G)}(p) \qquad G(z, 1) = 1, \quad (52)$$

where $G$ parametrises the usual gauge invariance. In the Heisenberg picture, $q_x(p)$ time evolves according to the Moyal equation

$$\boxed{i\hbar \partial_t q_{x,t}(p) = q_{x,t}(p) \star \varepsilon(p) - \varepsilon(p) \star q_{x,t}(p),} \quad (53)$$

which manifests the closure of the LQCOs under time evolution.

By definition, the density of a conserved operator is an LQCO.

We do not expect the existence of interacting (*i.e.*, not quadratic) charges, which, in turn, rules out the existence of interacting locally quasi-conserved operators; to the best of our knowledge, however, this has not been proved.

Finally, we remind the reader that the set of charges that are obtained by fixing the parameter $\kappa$ (in our case we have chosen $\kappa = 1$) is not always complete - this was originally pointed out in Ref. [100]. In particular, additional charges appear when the symbol of the Hamiltonian in some representation $\kappa$ is degenerate for generic momentum, which is possible when the dispersion relation in the $\kappa = 1$ representation has symmetries like $\varepsilon(p) = \pm\varepsilon(\pi \pm p)$.

**Charge densities.** A charge density can be defined as a locally quasi-conserved operator, associated with a given site, that is conserved when integrated over the position (*i.e.*, summed over all the sites and multiplied by $\mathfrak{a}$). By virtue of (64), it is natural to define its density as the quadratic form of fermions with the following symbol

$$\hat{q}_x^{(\ell)} = \mathfrak{a}^{-1} e^{-i\frac{\phi\left(p - \frac{i\hbar\partial_x}{2}\right)}{2}\sigma^y} e^{-i\frac{\theta\left(p - \frac{i\hbar\partial_x}{2}\right)}{2}\sigma^z} [q_o(p) + \sigma^y q_e(p)] e^{i\frac{\theta\left(p + \frac{i\hbar\partial_x}{2}\right)}{2}\sigma^z} e^{i\frac{\phi\left(p + \frac{i\hbar\partial_x}{2}\right)}{2}\sigma^y} \delta_{x,(\ell+s)\mathfrak{a}}, \quad (54)$$

where $s = 0, \frac{1}{2}$ is such that $q(p + \hbar\frac{\pi}{\mathfrak{a}}) = (-1)^{2s} q(p)$.

**Currents.** The current associated with a given charge can be easily defined from the Moyal dynamical equation (53), which can be rewritten as

$$\partial_t q_{x,t}^{(\ell)}(p) - 2\frac{e^{\mathfrak{a}\partial_x} - 1}{\hbar} \left[ \varepsilon(p) \frac{\sin\left(\hbar\frac{\overleftarrow{\partial_p}\overrightarrow{\partial_x}}{2}\right)}{e^{\mathfrak{a}\overrightarrow{\partial_x}} - 1} q_{x,t}^{(\ell)}(p) \right] = 0. \quad (55)$$

We can then use that, independently of the exact definition, the translational invariance of the charges implies the charge density $q_{x,t}^{(\ell)}(p)$ to be a function of $x - \ell\mathfrak{a}$. Thus we have

$$\partial_t q_{x,t}^{(\ell)}(p) + 2\frac{1 - e^{-\mathfrak{a}\partial_{\mathfrak{a}\ell}}}{\hbar} \left[ \varepsilon(p) \frac{\sin\left(\hbar\frac{\overleftarrow{\partial_p}\overrightarrow{\partial_x}}{2}\right)}{e^{\mathfrak{a}\overrightarrow{\partial_x}} - 1} q_{x,t}^{(\ell)}(p) \right] = 0, \quad (56)$$

*i.e.*

$$\partial_t q_{x,t}^{(\ell)}(p) + \frac{2}{\hbar}\varepsilon(p)\frac{\sin\left(\hbar\frac{\overleftarrow{\partial_p}\overrightarrow{\partial_x}}{2}\right)}{e^{\mathfrak{a}\overrightarrow{\partial_x}}-1}q_{x,t}^{(\ell)}(p) - \frac{2}{\hbar}\varepsilon(p)\frac{\sin\left(\hbar\frac{\overleftarrow{\partial_p}\overrightarrow{\partial_x}}{2}\right)}{e^{\mathfrak{a}\overrightarrow{\partial_x}}-1}q_{x,t}^{(\ell-1)}(p) = 0. \tag{57}$$

Finally, we can define the current as the LQCO with eigenvalue

$$j_x^{(\ell)}(p) = \frac{2\mathfrak{a}}{\hbar}\varepsilon(p)\frac{\sin\left(\hbar\frac{\overleftarrow{\partial_p}\overrightarrow{\partial_x}}{2}\right)}{e^{\mathfrak{a}\overrightarrow{\partial_x}}-1}q_{x,t}^{(\ell)}(p) = \int \mathrm{d}y \int \frac{\mathrm{d}k}{2\pi}\frac{\varepsilon\left(p+\hbar\frac{k}{\mathfrak{a}}\right)-\varepsilon\left(p-\hbar\frac{k}{\mathfrak{a}}\right)}{\hbar\sin k}e^{2i\frac{k(x-y)}{\mathfrak{a}}}q_{y-\frac{\mathfrak{a}}{2},t}^{(\ell)}(p) =$$

$$\mathfrak{a}\sum_n\int_{-\pi}^{\pi}\frac{\mathrm{d}k}{2\pi}\frac{\varepsilon\left(p+\hbar\frac{k}{\mathfrak{a}}\right)-\varepsilon\left(p-\hbar\frac{k}{\mathfrak{a}}\right)}{2\hbar\sin k}e^{i(n+1)k}q_{x+n\frac{\mathfrak{a}}{2},t}^{(\ell)}(p). \tag{58}$$

By integrating over the position $\ell\mathfrak{a}$, we find the eigenvalue of the total current $j(p) = v(p)q(p)$.

We note that such definition is different from the one exhibited in [45], where the charge densities were defined in a different way (in particular, they were chosen to be local).

### 3.2.5 Off-diagonal operators

Similarly to what we did for states, we also identify another class of quadratic operators, which we call "off-diagonal operators", that is invariant under time evolution. They have the symbol

$$\hat{b}_x^{\mathrm{ODO}}(z_p; G) = e^{-i\frac{\phi\left(p-\frac{i\hbar\overrightarrow{\partial_x}}{2}\right)}{2}\sigma^y}e^{-i\frac{\theta\left(p-\frac{i\hbar\overrightarrow{\partial_x}}{2}\right)}{2}\sigma^z}\left\{\sigma^z b_{x,\mathrm{R}}^{(G)}(p) - \sigma^x b_{x,\mathrm{I}}^{(G)}(p)\right\}e^{i\frac{\theta\left(p+\frac{i\hbar\overleftarrow{\partial_x}}{2}\right)}{2}\sigma^z}e^{i\frac{\phi\left(p+\frac{i\hbar\overleftarrow{\partial_x}}{2}\right)}{2}\sigma^y}. \tag{59}$$

In the Heisenberg picture, $b_x(p)$ time evolves as follows

$$i\hbar\partial_t b_{x,t}(p) = \varepsilon(p)\star b_{x,t}(-p) - b_{x,t}(p)\star\varepsilon(-p). \tag{60}$$

### 3.3 Expectation values

Formula (23) can be written in an explicitly symmetrical form:

$$\langle\boldsymbol{O}\rangle_t = \sum_{\frac{x}{\mathfrak{a}}\in\mathbb{Z}}\int_{-\pi}^{\pi}\frac{\mathrm{d}[\mathfrak{a}p/\hbar]}{8\pi\kappa}\mathrm{tr}[\hat{\Gamma}_{x,t}(z_p)\star\hat{o}_x(z_p)]. \tag{61}$$

For $\kappa = 1$, by decomposing a quasi-localised quadratic operator in its quasi-conserved and off-diagonal parts, we can improve the parametrisation (26) as follows:

$$\hat{o}(z_p, w) = e^{-i\frac{\phi\left(p+i\frac{w}{2}\right)}{2}\sigma^y}e^{-i\frac{\theta\left(p+i\frac{w}{2}\right)}{2}\sigma^z}\left[\omega(p,w)\frac{\mathrm{I}+\sigma^y}{2} - \omega(-p,w)\frac{\mathrm{I}-\sigma^y}{2} + \right.$$

$$\left.\left(\Omega_{o_1}(p,w)\frac{\sigma^z-i\sigma^x}{2} + h.c.\right)\right]e^{i\frac{\theta\left(p-i\frac{w}{2}\right)}{2}\sigma^z}e^{i\frac{\phi\left(p-i\frac{w}{2}\right)}{2}\sigma^y}. \tag{62}$$

In this way, formula (27) still holds, provided to replace spuriously local root density and auxiliary field by their genuinely local counterparts.

**Locally quasi-conserved operators.** The expectation value of an LQCO in a gaussian state with correlation matrix as in (41) takes a rather simple form (in the standard gauge $G = 1$):

$$\langle\boldsymbol{Q}\rangle = \frac{\mathfrak{a}}{2}\sum_{\frac{x}{\mathfrak{a}}\in\frac{1}{2}\mathbb{Z}}\int_{-\hbar\frac{\pi}{\mathfrak{a}}}^{\hbar\frac{\pi}{\mathfrak{a}}}\mathrm{d}p\left[\rho_x(p) - \frac{1}{4\pi\hbar}\right]\left[q_x(p) + (-1)^{2\frac{x}{\mathfrak{a}}}q_x\left(p+\hbar\frac{\pi}{\mathfrak{a}}\right)\right]. \tag{63}$$

It is then convenient to classify the LQCOs depending on how their eigenvalues transform under $p \to p + \hbar\frac{\pi}{\mathfrak{a}}$:

$$
\langle Q \rangle =
\begin{cases}
\mathfrak{a}\sum_{\frac{x}{\mathfrak{a}}\in\mathbb{Z}}\int_{-\hbar\frac{\pi}{\mathfrak{a}}}^{\hbar\frac{\pi}{\mathfrak{a}}} dp[\rho_x(p) - \frac{1}{4\pi\hbar}]q_x(p) & q_x(p + \hbar\frac{\pi}{\mathfrak{a}}) = q_x(p) \\
\mathfrak{a}\sum_{\frac{x}{\mathfrak{a}}\in\frac{1}{2}+\mathbb{Z}}\int_{-\hbar\frac{\pi}{\mathfrak{a}}}^{\hbar\frac{\pi}{\mathfrak{a}}} dp\,\rho_x(p)q_x(p) & q_x(p + \hbar\frac{\pi}{\mathfrak{a}}) = -q_x(p).
\end{cases}
\tag{64}
$$

**Charge densities.** Using Ansatz (54), the expectation value of a generic charge density reads as

$$
\langle Q_\ell \rangle = \sum_{s=0,\frac{1}{2}} \int_{-\hbar\frac{\pi}{\mathfrak{a}}}^{\hbar\frac{\pi}{\mathfrak{a}}} dp \left[ \rho_{(\ell+s)\mathfrak{a}}(p) - \frac{1}{4\pi\hbar} \right] \frac{q(p) + (-1)^{2s}q(p + \hbar\frac{\pi}{\mathfrak{a}})}{2}.
\tag{65}
$$

### 3.4 Inhomogeneous Hamiltonians

In this section we extend the previous analysis to inhomogeneous Hamiltonians. If the symbol of the correlation matrix is parametrised as in (41), it could be convenient to express the symbol of the Hamiltonian as follows

$$
\hat{h}_x(z_p) = e^{-i\frac{\phi\left(p - \frac{i\hbar\overrightarrow{\partial_x}}{2}\right)}{2}\sigma^y} e^{-i\frac{\theta\left(p - \frac{i\hbar\overrightarrow{\partial_x}}{2}\right)}{2}\sigma^z} \times
$$

$$
\left\{ \varepsilon_{x,o}(p)\mathrm{I} + \varepsilon_{x,e}(p)\sigma^y + w_{x,\mathrm{R}}(p)\sigma^z - w_{x,\mathrm{I}}(p)\sigma^x \right\} e^{i\frac{\theta\left(p + \frac{i\hbar\overleftarrow{\partial_x}}{2}\right)}{2}\sigma^z} e^{i\frac{\phi\left(p + \frac{i\hbar\overleftarrow{\partial_x}}{2}\right)}{2}\sigma^y}, \tag{66}
$$

where $w_x(p) = -w_x(-p)$ is an odd complex field.

Eq. (66) generates the following dynamics

$$
i\hbar\partial_t\rho_{x,t}(p) = \varepsilon_x(p) \star \rho_{x,t}(p) - \rho_{x,t}(p) \star \varepsilon_x(p) + w_x(p) \star \Psi_{x,t}^*(p) - \Psi_{x,t}(p) \star w_x^*(p)
\tag{67}
$$

$$
i\hbar\partial_t\Psi_{x,t}(p) = \varepsilon_x(p) \star \Psi_{x,t}(p) + \Psi_{x,t}(p) \star \varepsilon_x(-p) - w_x(p) \star \rho_{x,t}(-p) - \rho_{x,t}(p) \star w_x(p).
\tag{68}
$$

This parametrisation clearly manifests that the two invariant subspaces identified before for homogeneous Hamiltonians, *i.e.*, the LQSS and the ODS, are invariant even when time evolution is generated by a locally quasi-conserved operator ($w_x(p) = 0$).

We mention that, if time evolution is generated by an off-diagonal operator ($\varepsilon_x(p) = 0$), the dynamics can still be written in a decoupled form

$$
\begin{aligned}
-\hbar^2\partial_t^2\rho_{x,t}(p) &= [w_x(p) \star w_x^*(p)] \star \rho_{x,t}(p) + 2w_x(p) \star \rho_{x,t}(-p) \star w_x^*(p) + \\
&\qquad + \rho_{x,t}(p) \star [w_x(p) \star w_x^*(p)] \\
-\hbar^2\partial_t^2\Psi_{x,t}(p) &= [w_x(p) \star w_x^*(p)] \star \Psi_{x,t}(p) + 2w_x(p) \star \Psi_{x,t}^*(p) \star w_x(p) + \\
&\qquad + \Psi_{x,t}(p) \star [w_x^*(p) \star w_x(p)]
\end{aligned}
\tag{69}
$$

with the additional (coupled) boundary conditions

$$
\begin{aligned}
i\hbar\partial_t\rho_{x,t}(p)\Big|_{t=0} &= w_x(p) \star \Psi_{x,0}^*(p) - \Psi_{x,0}(p) \star w_x^*(p) \\
i\hbar\partial_t\Psi_{x,t}(p)\Big|_{t=0} &= -w_x(p) \star \rho_{x,0}(-p) - \rho_{x,0}(p) \star w_x(p).
\end{aligned}
\tag{70}
$$

Equation (66) is a sensible parametrisation when $w_x(p)$ can be treated as a small perturbation with respect to $\varepsilon_x(p)$, for example assuming $w_x(p) = -i\hbar\partial_x c_x(p)$, with $c_x(p)$ a weakly inhomogeneous odd complex field with the dimension of a velocity. Such term could arise

from a naive definition of an LQCO - for example, overlooking the derivatives in (51). At the first order in the inhomogeneity we then have

$$\partial_t \rho = -\partial_x[v\rho] + \partial_p[(\partial_x \varepsilon)\rho] + \text{Re}[(\partial_x c)\Psi^*] + O(\partial_x^2)$$

$$\partial_t \Psi + i\frac{2\varepsilon_e}{\hbar}\Psi = -\partial_x[v_e \Psi] + \partial_p[(\partial_x \varepsilon_o)\Psi] + 2(\partial_x c)\rho_e + O(\partial_x^2), \tag{71}$$

where the arguments of the functions ($x$, $t$ and $p$) are understood.

In the most generic case, it is instead convenient to express the symbol of the Hamiltonian as follows:

$$\hat{h}_x(z_p) = e_\star^{-i\frac{\hat{\Theta}_x(z_p)}{2}} \star \left[\varepsilon_{x,o}(p)\text{I} + \varepsilon_{x,e}(p)\sigma^y\right] \star e_\star^{i\frac{\hat{\Theta}_x(z_p)}{2}}, \tag{72}$$

where $\hat{\Theta}_x(z_p)$ is a two-by-two Hermitian matrix that generalises the Bogoliubov angle and satisfies $\hat{\Theta}_x(1/z_p) = -\hat{\Theta}_x^t(z_p)$. The correlation matrix can then be parametrised as

$$\hat{\Gamma}_x(z_p) = 4\pi\hbar e_\star^{-i\frac{\hat{\Theta}_x(z_p)}{2}} \star \left\{\text{I}\rho_{x,o}(p) + \sigma^y\left[\rho_{x,e}(p) - \frac{1}{4\pi\hbar}\right] + \sigma^z\Psi_{x,\text{R}}(p) - \sigma^x\Psi_{x,\text{I}}(p)\right\} \star e_\star^{i\frac{\hat{\Theta}_x(z_p)}{2}}. \tag{73}$$

In this way the root density and the auxiliary field satisfy a simple generalisation of (42), (43)

$$i\hbar\partial_t\rho_{x,t}(p) = \varepsilon_x(p) \star \rho_{x,t}(p) - \rho_{x,t}(p) \star \varepsilon_x(p) \tag{74}$$

$$i\hbar\partial_t\Psi_{x,t}(p) = -\varepsilon_x(p) \star \Psi_{x,t}(-p) + \Psi_{x,t}(p) \star \varepsilon_x(-p). \tag{75}$$

**Proof of** (74) **and** (75). The equations of motions with the parametrisation (72) can be easily worked out by observing that the Bogoliubov transformation is unitary and, for any $\hat{U}(x,p)$ such that $\hat{U}(x,p) \star \hat{U}^\dagger(x,p) = \text{I}$, we have

$$[\hat{U}(x,p) \star \hat{f}(x,p) \star \hat{U}^\dagger(x,p)] \star [\hat{U}(x,p) \star \hat{g}(x,p) \star \hat{U}^\dagger(x,p)] =$$
$$\hat{U}(x,p) \star \hat{f}(x,p) \star [\hat{U}^\dagger(x,p) \star \hat{U}(x,p)] \star \hat{g}(x,p) \star \hat{U}^\dagger(x,p) =$$
$$\hat{U}(x,p) \star \hat{f}(x,p) \star \hat{g}(x,p) \star \hat{U}^\dagger(x,p). \tag{76}$$

This allows us to reduce the star products between the symbols in (31) to star products between the functions ($\rho, \Psi, \varepsilon$) characterising the symbols, and finally to get (74) and (75). In the specific case of homogeneous Hamiltonians, the Bogoliubov transformation is independent of $x$, i.e., $\hat{U}(x,p) = \hat{U}(p)$ in (76), and we can use $\hat{U}(p)\star\hat{f}(x,p)\star\hat{U}^\dagger(p) = \hat{U}\left(p - \frac{i\hbar\overrightarrow{\partial_x}}{2}\right)\hat{f}(x,p)\hat{U}^\dagger\left(p + \frac{i\hbar\overleftarrow{\partial_x}}{2}\right)$, which explains (41) and analogues expressions.

### 3.4.1 Integrals of motion

In the inhomogeneous case we can immediately identify a class of commuting quadratic operators, namely, the LQCOs with eigenvalues of the form

$$q_x(p) = Q_\star[\varepsilon_x(p)], \tag{77}$$

where $Q_\star$ is a generic $\star$-function, i.e., a funcion in which multiplications are replaced by Moyal star products. This class is not expect to be complete, especially if the dispersion relation has particular symmetries. For example, if $\varepsilon_x(p) = \varepsilon_x(-p)$, all the charges of the form (77) are even; this contrasts with the homogeneous case where there are infinitely many odd charges (under spatial reflections).

## 3.5 More general invariant subspaces

We still assume $\kappa = 1$ and consider time evolution under a locally quasi-conserved operator.

From (67) we can easily infer that there are also invariant subspaces with both root density and complex field different from zero. Specifically, the states lying on surfaces of the form

$$\mathcal{P}_\star(\rho_x(p), \Psi_x(p) \star \Psi_x^*(p), \Psi_x^*(p) \star \Psi_x(p), \varepsilon_x(p)) = \text{const}, \tag{78}$$

never leave the surfaces. For example, the following state belongs to the invariant subspace on the surface $\rho_x(p) = \frac{\beta}{\alpha^2} \Psi_x(p) \star \Psi_x^*(p)$:

$$
\begin{aligned}
\Psi_x(p) =& \alpha \sin\left(\frac{kx}{\hbar}\right) \sin\left(\frac{n\mathfrak{a}p}{\hbar}\right) \\
\rho_x(p) =& \beta \left[\cos^2\left(\frac{n\mathfrak{a}k}{2\hbar}\right) \sin^2\left(\frac{kx}{\hbar}\right) \sin^2\left(\frac{n\mathfrak{a}p}{\hbar}\right) + \sin^2\left(\frac{n\mathfrak{a}k}{2\hbar}\right) \cos^2\left(\frac{kx}{\hbar}\right) \cos^2\left(\frac{n\mathfrak{a}p}{\hbar}\right)\right].
\end{aligned}
\tag{79}
$$

Finally, we note that, if the Hamiltonian is homogeneous, the constant in (78) can be replaced by any function of the momentum.

## 4 The two-temperature scenario revisited

In this section we reconsider the two-temperature scenario, which Ref. [61] studied extensively in the transverse-field Ising chain. There, the initial state was the junction of two states prepared at different temperatures in two semi-infinite open chains. Perhaps counterintuitively, that is not an LQSS, and hence the corrections are not purely hydrodynamic. We would like to choose a locally quasi-stationary initial state with a step root density; this would however be inconsistent with our original hypothesis of smoothness of the symbol. Thus, we replace the step function by a smooth function that is equivalent to a step function for $\frac{x}{\mathfrak{a}} \in \frac{1}{2}\mathbb{Z}$, with a given convention on $\text{sgn}(0) \in \{-1, 0, 1\}$, as follows

$$
\rho_{x,0}(p) = \frac{1}{2\pi\hbar} \frac{1}{1 + e^{\beta_L \varepsilon(p)}} + \frac{\frac{1}{1+e^{\beta_R \varepsilon(p)}} - \frac{1}{1+e^{\beta_L \varepsilon(p)}}}{2\pi\hbar} \lim_{\epsilon \to 0} \int_{-\pi}^{\pi} \frac{dq}{4\pi} e^{\frac{2iqx}{\mathfrak{a}}} \left[\text{sgn}(0) - i \cot\frac{q - i\epsilon}{2}\right] \tag{80}
$$
$$\Psi_{x,0}(p) = 0.$$

The solution to the dynamics is given by (45)

$$
\begin{aligned}
\rho_{x,t}(p) =& \frac{1}{2\pi\hbar} \frac{1}{1 + e^{\beta_L \varepsilon(p)}} + \frac{1}{2\pi\hbar} \left[\frac{1}{1 + e^{\beta_R \varepsilon(p)}} - \frac{1}{1 + e^{\beta_L \varepsilon(p)}}\right] \times \\
& \lim_{\epsilon \to 0} \int_{-\pi}^{\pi} \frac{dq}{4\pi} e^{\frac{2iqx}{\mathfrak{a}}} e^{-it \frac{\varepsilon\left(p + \frac{\hbar}{\mathfrak{a}}q\right) - \varepsilon\left(p - \frac{\hbar}{\mathfrak{a}}q\right)}{\hbar}} \left[\text{sgn}(0) - i \cot\frac{q - i\epsilon}{2}\right]
\end{aligned}
\tag{81}
$$
$$\Psi_{x,t}(p) = 0.$$

From this expression we can readily identify the entire correction to first-order GHD:

$$
\begin{aligned}
\partial_t \rho_{x,t}(p) + v(p)\partial_x \rho_{x,t}(p) =& \frac{1}{2\pi\hbar} \left[\frac{1}{1 + e^{\beta_L \varepsilon(p)}} - \frac{1}{1 + e^{\beta_R \varepsilon(p)}}\right] \times \\
& \int_{-\pi}^{\pi} \frac{dq}{2\pi\hbar} e^{\frac{2iqx}{\mathfrak{a}}} e^{-it \frac{\varepsilon\left(p + \frac{\hbar}{\mathfrak{a}}q\right) - \varepsilon\left(p - \frac{\hbar}{\mathfrak{a}}q\right)}{\hbar}} \frac{\varepsilon\left(p + \frac{\hbar}{\mathfrak{a}}q\right) - \varepsilon\left(p - \frac{\hbar}{\mathfrak{a}}q\right) - 2\frac{\hbar}{\mathfrak{a}}qv(p)}{2q} \left[q \cot\frac{q}{2} + iq\,\text{sgn}(0)\right],
\end{aligned}
\tag{82}
$$

which generally approaches zero as $t^{-\frac{1}{2}}$ in a rapidly oscillating way (for $v(p) = \frac{x}{t}$, it generally does as $t^{-1}$). We remind the reader that the expectation values of local observables have a

further integral over the momentum (*cf.* (61)), therefore, due to the rapidly oscillating terms, the corrections to the first-order result tend to decay to zero faster, generally as $t^{-1}$.

For the sake of simplicity, we write down the symbol of the correlation matrix only for reflection symmetric ($\varepsilon(p) = \varepsilon(-p)$) generalised XY Hamiltonians with $\phi(p) = 0$

$$
\hat{\Gamma}^{(\beta_L,\beta_R)}_{x,t}(z_p) = -\tanh\left(\frac{\beta_L \varepsilon(p)}{2}\right)\sigma^y e^{i\theta(p)\sigma^z} + \left[\tanh\left(\frac{\beta_L \varepsilon(p)}{2}\right) - \tanh\left(\frac{\beta_R \varepsilon(p)}{2}\right)\right] \times
$$

$$
\lim_{\epsilon \to 0}\int_{-\pi}^{\pi}\frac{\mathrm{d}q}{4\pi}e^{\frac{2iqx}{\mathfrak{a}}}\left[\mathrm{sgn}(0) - i\cot\frac{q - i\epsilon}{2}\right]\left[\cos\left(t\frac{\varepsilon\left(p+\frac{\hbar}{\mathfrak{a}}q\right) - \varepsilon\left(p-\frac{\hbar}{\mathfrak{a}}q\right)}{\hbar}\right)\sigma^y e^{i\frac{\theta\left(p-\frac{\hbar}{\mathfrak{a}}q\right)+\theta\left(p+\frac{\hbar}{\mathfrak{a}}q\right)}{2}\sigma^z} - \right.
$$

$$
\left. i\sin\left(t\frac{\varepsilon\left(p+\frac{\hbar}{\mathfrak{a}}q\right) - \varepsilon\left(p-\frac{\hbar}{\mathfrak{a}}q\right)}{\hbar}\right)e^{i\frac{\theta\left(p-\frac{\hbar}{\mathfrak{a}}q\right)-\theta\left(p+\frac{\hbar}{\mathfrak{a}}q\right)}{2}\sigma^z}\right]. \quad (83)
$$

We point out that (83) is not always physical, as some eigenvalues of the corresponding correlation matrix $\Gamma$ could be outside the physical interval $[-1,1]$. In that case, an LQSS with a step root density does not exist, *i.e.*, that state can not be prepared. In the marginal case the correlation matrix has an eigenvalue exactly equal to $\pm 1$, and the step profile is an eigenstate of some charge.

## 4.1 Example (reduction to finite chains)

In this subsection we provide a numerical check of our prediction for the time evolution of LQSSs. In order to obtain numerical data by standard means, we are forced to consider finite chains, so we must adapt our description for systems with a finite number of spins. We propose to embed a finite periodic fermionic chain into an infinite one. The first step is to impose periodicity in space $\rho_{x+L\mathfrak{a},0}(p) = \rho_{x,0}(p)$, $\Psi_{x+L\mathfrak{a},0}(p) = \Psi_{x,0}(p)$. For example, (80) could be modified as follows:

$$
\rho_{x,0}(p) = \frac{\frac{1}{1+e^{\beta_R \varepsilon(p)}} + \frac{1}{1+e^{\beta_L \varepsilon(p)}}}{4\pi\hbar} - \frac{\frac{1}{1+e^{\beta_R \varepsilon(p)}} - \frac{1}{1+e^{\beta_L \varepsilon(p)}}}{4\pi\hbar L}\sum_{n=1}^{L}\frac{\sin\left(\frac{(4gx-\mathfrak{a})(2n-1)\pi}{2L\mathfrak{a}}\right)}{\sin\left(\frac{(2n-1)\pi}{2L}\right)} \quad (84)
$$

$$
\Psi_{x,0}(p) = 0\,,
$$

where the chain is split in $2g$ parts with alternate temperatures $\{\beta_L, \beta_R, \beta_L, \ldots\}$; the parts have the same length if $L$ is divisible by $2g$. This is not the end of the story: periodicity in space is not sufficient to describe a finite periodic fermionic chain, indeed in that case the momentum is not a continuous variable as in (84) but it is quantised according to $e^{i\frac{L\mathfrak{a}p}{\hbar}} = 1$. This problem can be circumvented by observing that the Moyal star product (32) between any translationally invariant, $2\pi$-periodic function of $\frac{\mathfrak{a}p}{\hbar}$ and the root density (84) produces terms where the momentum is shifted by $\frac{gn\pi}{L}\frac{\hbar}{\mathfrak{a}}$, with $n$ odd; thus, if $g$ is even, the Moyal star product turns out to preserve the Ramond quantisation condition $e^{i\frac{L\mathfrak{a}p}{\hbar}} = 1$. In conclusion, for even $g$, time evolution in the finite fermionic chain with periodic boundary conditions can be obtained from our results valid in the thermodynamic limit, provided to impose the Ramond quantisation conditions and to replace integrals over momenta with sums. Specifically, the time dependent root density reads

$$
\rho_{x,t}(p) = \frac{\frac{1}{1+e^{\beta_R \varepsilon(p)}} + \frac{1}{1+e^{\beta_L \varepsilon(p)}}}{4\pi\hbar} -
$$

$$
\frac{\frac{1}{1+e^{\beta_R \varepsilon(p)}} - \frac{1}{1+e^{\beta_L \varepsilon(p)}}}{4\pi\hbar L}\sum_{n=1}^{L}\frac{\sin\left(\frac{(4gx-\mathfrak{a})(2n-1)\pi}{2L\mathfrak{a}} - t\frac{\varepsilon\left(p+\hbar\frac{(2n-1)g\pi}{L\mathfrak{a}}\right) - \varepsilon\left(p-\hbar\frac{(2n-1)g\pi}{L\mathfrak{a}}\right)}{\hbar}\right)}{\sin\left(\frac{(2n-1)\pi}{2L}\right)}. \quad (85)
$$

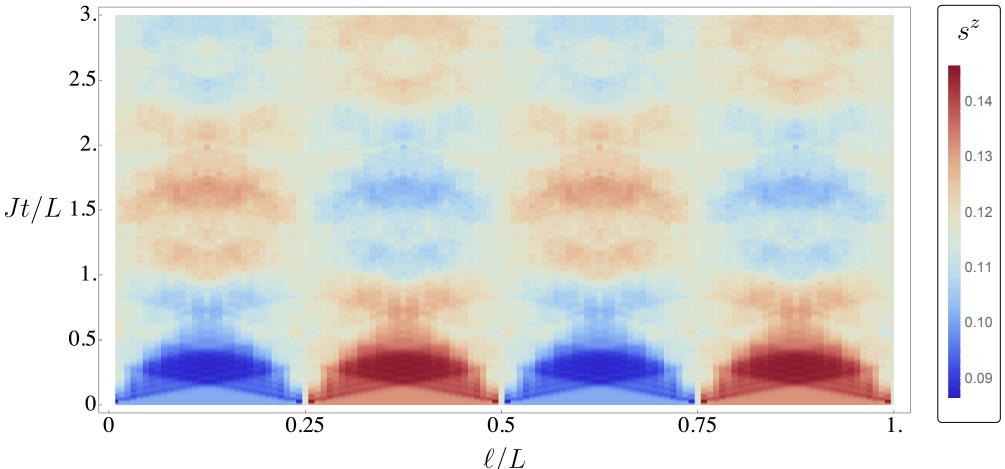

Figure 1: The profile of the local magnetisation $s_\ell^z(t)$ in a chain with $L = 128$ spins with periodic boundary conditions on the Jordan-Wigner fermions. The system is prepared in the 4-partite LQSS at temperatures $\{\beta_L, \beta_R, \beta_L, \beta_R\}$, with $\beta_L = J^{-1}$ and $\beta_R = 2J^{-1}$ ($g = 2$ in (87)). Time evolution is generated by the Hamiltonian $H = J \sum_{\ell=1}^{L}[\frac{1}{8}\sigma_\ell^y \sigma_{\ell+1}^y - \frac{5}{8}\sigma_\ell^x \sigma_{\ell+1}^x - \frac{1}{4}\sigma_\ell^z]$ (with periodic boundary conditions on the J-W fermions).

For the sake of example, we report the expression for the magnetisation profile

$$s_\ell^z(t) = \frac{1}{2}\langle \sigma_\ell^z \rangle_t = \frac{i}{2L} \sum_{n=1}^{L} \left[\hat{\Gamma}_{\ell,t}(e^{\frac{2\pi i n}{L}})\right]_{12}, \tag{86}$$

which can be written as follows

$$s_\ell^z(t) = -\frac{1}{4L} \sum_{k \in R}\left\{ \frac{\sinh\left(\frac{\beta_L + \beta_R}{2}\varepsilon_k\right)}{\cosh\left(\frac{\beta_R}{2}\varepsilon_k\right)\cosh\left(\frac{\beta_L}{2}\varepsilon_k\right)}\cos\theta_k + \frac{\sinh\left(\frac{\beta_L - \beta_R}{2}\varepsilon_k\right)}{\cosh\left(\frac{\beta_R}{2}\varepsilon_k\right)\cosh\left(\frac{\beta_L}{2}\varepsilon_k\right)}\frac{1}{L}\sum_{n=1}^{L}\right.$$

$$\left. \frac{\sin\left[(4g\ell - 1)\frac{2n-1}{2L}\pi\right]\cos\left[\frac{\theta_{k+(2n-1)\frac{g\pi}{L}} + \theta_{k-(2n-1)\frac{g\pi}{L}}}{2}\right]\cos\left[(\varepsilon_{k+(2n-1)\frac{g\pi}{L}} - \varepsilon_{k-(2n-1)\frac{g\pi}{L}})t\right]}{\sin\left(\frac{2n-1}{2L}\pi\right)}\right\}, \tag{87}$$

where $\theta_k = \theta(\hbar k/\mathfrak{a})$, $\varepsilon_k = \hbar^{-1}\varepsilon(\hbar k/\mathfrak{a})$, and R is the set of all the independent solutions $k$ to $e^{iLk} = 1$. We warn the reader that, in order to use (87), the Bogoliubov angle must be (defined as) a smooth odd function of $k$.

We have checked that, for even $g$, (87) is exactly equal to the local magnetisation in the state obtained by time evolving the initial correlation matrix, which has the symbol

$$\hat{\Gamma}_{x,0}(e^{ik}) = -\frac{1}{2}e^{-i\theta(p)\sigma^z}\sigma^y\left[\tanh\left(\frac{\beta_R \varepsilon_k}{2}\right) + \tanh\left(\frac{\beta_L \varepsilon_k}{2}\right)\right] - \frac{1}{2}\left[\tanh\left(\frac{\beta_R \varepsilon_k}{2}\right) - \tanh\left(\frac{\beta_L \varepsilon_k}{2}\right)\right]\times$$

$$\frac{1}{L}\sum_{n=1}^{L}e^{-i\frac{\theta\left(p + \frac{\hbar g(2n-1)\pi}{L\mathfrak{a}}\right) + \theta\left(p - \frac{\hbar g(2n-1)\pi}{L\mathfrak{a}}\right)}{2}}\sigma^z\sigma^y\frac{\sin\left(\frac{(4gx-\mathfrak{a})(2n-1)\pi}{2L\mathfrak{a}}\right)}{\sin\left(\frac{(2n-1)\pi}{2L}\right)} \quad (e^{iLk} = 1). \tag{88}$$

Figure 1 shows $s_\ell^z(t)$ when the system is prepared in a 4-partite LQSS[4].

---

[4] The numerical data have been obtained by evaluating (87), which is more efficient than explicitly time evolving the initial correlation matrix.

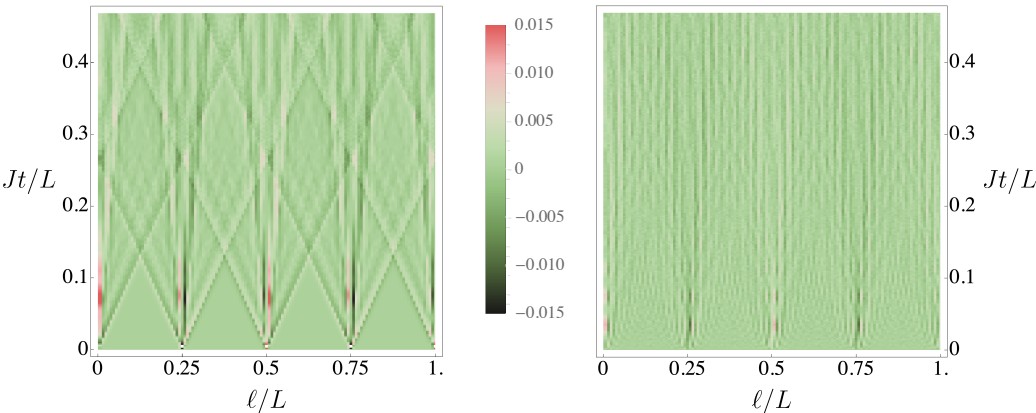

Figure 2: The error made in the calculation of $s^z$ when using generalised hydrodynamics at the first (left panel) and at the third (right panel) order for the dynamics of fig. 1.

We conclude with a curiosity, readily recognisable by inspecting (87): despite the total spin in the $z$ direction not being generally conserved, its expectation value $S^z = \sum_\ell s^z_\ell$ is stationary. This phenomenon does not actually depend on the observable, but it is a consequence of the state being locally quasi-stationary.

**Asymptotic expansion.** It can be instructive to carry out the asymptotic expansion of (85), which will give the solution to the generalised hydrodynamic equation at a given order in the inhomogeneity (*cf.* (44)). We will assume $L$ divisible by 2. Using the parity of the function in the sum, we can rewrite (85) as follows:

$$
\rho_{x,t}(p) = \frac{\frac{1}{1+e^{\beta_R \varepsilon(p)}} + \frac{1}{1+e^{\beta_L \varepsilon(p)}}}{4\pi\hbar} -
$$
$$
\frac{\frac{1}{1+e^{\beta_R \varepsilon(p)}} - \frac{1}{1+e^{\beta_L \varepsilon(p)}}}{2\pi\hbar L} \sum_{n=1}^{\frac{L}{2}} \frac{\sin\left(\frac{(4gx-\mathfrak{a})(2n-1)\pi}{2L\mathfrak{a}} - t\frac{\varepsilon\left(p+\hbar\frac{(2n-1)g\pi}{L\mathfrak{a}}\right)-\varepsilon\left(p-\hbar\frac{(2n-1)g\pi}{L\mathfrak{a}}\right)}{\hbar}\right)}{\sin\left(\frac{(2n-1)\pi}{2L}\right)}. \quad (89)
$$

First-order hydrodynamics predicts $\rho^{(1)}_{x,t}(p) = \rho_{x-v(p)t,0}(p)$, that is to say

$$
\rho^{(1)}_{x,t}(p) = \frac{\frac{1}{1+e^{\beta_R \varepsilon(p)}} + \frac{1}{1+e^{\beta_L \varepsilon(p)}}}{4\pi\hbar} -
$$
$$
\frac{\frac{1}{1+e^{\beta_R \varepsilon(p)}} - \frac{1}{1+e^{\beta_L \varepsilon(p)}}}{2\pi\hbar L} \sum_{n=1}^{\frac{L}{2}} \frac{\sin\left(\frac{(4gx-\mathfrak{a})(2n-1)\pi}{2L\mathfrak{a}} - t\frac{2g(2n-1)\pi v(p)}{L\mathfrak{a}}\right)}{\sin\left(\frac{(2n-1)\pi}{2L}\right)}. \quad (90)
$$

The solution to third-order hydrodynamics is $\rho_{x,t}^{(3)}(p) = \int_{-\infty}^{\infty} dy\,\mathrm{Ai}(y)\rho_{x-v(p)t+\frac{y}{2}\sqrt[3]{\hbar^2 v''(p)t},0}(p)$, which results in

$$
\rho_{x,t}^{(3)}(p) = \frac{\frac{1}{1+e^{\beta_R \varepsilon(p)}} + \frac{1}{1+e^{\beta_L \varepsilon(p)}}}{4\pi\hbar} -
$$
$$
\frac{\frac{1}{1+e^{\beta_R \varepsilon(p)}} - \frac{1}{1+e^{\beta_L \varepsilon(p)}}}{2\pi\hbar L} \sum_{n=1}^{\frac{L}{2}} \frac{\sin\left( \frac{(4gx-\mathfrak{a})(2n-1)\pi}{2L\mathfrak{a}} - t\frac{2g(2n-1)\pi v(p)}{L\mathfrak{a}} - t\frac{g^3(2n-1)^2\pi^3\hbar^2 v''(p)}{3L^3\mathfrak{a}^3} \right)}{\sin\left( \frac{(2n-1)g\pi}{2L} \right)} . \quad (91)
$$

An attentive reader might have noted that the expansion in the strength of the inhomogeneity is an expansion about $p$ of the argument of the dispersion relations that are multiplied by the time in (89). Appendix B shows that this is indeed general. We can then use this prescription in (87) (with the sum restricted to $\{1, \ldots, \frac{L}{2}\}$ and multiplied by 2, as in (89)) and check the goodness of higher-order hydrodynamics. Fig. 2 shows that the error of the approximation decreases as the order of GHD is increased. In addition, while the light-cones are clearly visible in the left panel, they are imperceptible in the right panel, showing that third-order hydrodynamics describes the behaviour around the light cones, as conjectured in Ref. [84].

# 5 Continuum scaling limit

So far we have considered a spin-chain system at fixed lattice spacing $\mathfrak{a}$, providing also perturbative expansions valid when the typical length of the inhomogeneity is much larger than $\mathfrak{a}$. In this section we adopt the alternative point of view of keeping the typical length of the inhomogeneity fixed while taking the limit $\mathfrak{a} \to 0$. The coupling constants can then be chosen to scale with $\mathfrak{a}$ in such a way to allow for a quantum field theory description of the low-energy degrees of freedom. This is a standard problem, but since our representation in terms of symbols could be unfamiliar to the reader, we describe the procedure in detail.

A preliminary step is to rescale the Majorana operators. We define the continuum Majorana fields as $\psi_i(x) = \lim_{\mathfrak{a}\to 0} \psi_i(x;\mathfrak{a})$, where

$$
\psi_i(x;\mathfrak{a}) = \frac{1}{\sqrt{\mathfrak{a}}} \sum_{\frac{y}{\mathfrak{a}}\in\mathbb{Z}} \frac{\sin\left(\pi\frac{x-y}{\mathfrak{a}}\right)}{\pi\frac{x-y}{\mathfrak{a}}} a_{2\kappa\frac{y}{\mathfrak{a}}+i}, \qquad \lim_{\mathfrak{a}\to 0}[\psi_i(x;\mathfrak{a}),\psi_j(y;\mathfrak{a})]_+ = 2\delta_{ij}\delta(x-y), \quad (92)
$$

and we used the sinc interpolation to extend the definition of the operators to $\mathbb{R}$; in this way the operators are entire functions of the position even when $\mathfrak{a}$ is finite (but they satisfy a more complicated algebra). In the following we use the vector notation $\vec{\psi}(x;\mathfrak{a}) = \begin{bmatrix} \psi_1(x;\mathfrak{a}) \\ \psi_2(x;\mathfrak{a}) \end{bmatrix}$. The Hamiltonian can be written as

$$
H = \frac{\mathfrak{a}^2}{4} \sum_{\frac{x}{\mathfrak{a}},\frac{y}{\mathfrak{a}}\in\frac{1}{2}\mathbb{Z}} \int_{-\hbar\frac{\pi}{\mathfrak{a}}}^{\hbar\frac{\pi}{\mathfrak{a}}} \frac{dp}{2\pi\hbar} e^{\frac{2i(x-y)p}{\hbar}} \vec{\psi}^\dagger(2x;\mathfrak{a})\hat{h}_{x+y}(e^{i\mathfrak{a}p/\hbar})\vec{\psi}(2y;\mathfrak{a}). \quad (93)
$$

By expanding the Majorana fields around $x + y$, repeated integrations by parts result in

$$
H = \frac{\mathfrak{a}^2}{4} \sum_{\frac{x}{\mathfrak{a}},\frac{y}{\mathfrak{a}}\in\frac{1}{2}\mathbb{Z}} \int_{-\hbar\frac{\pi}{\mathfrak{a}}}^{\hbar\frac{\pi}{\mathfrak{a}}} \frac{dp}{2\pi\hbar} e^{\frac{2i(x-y)p}{\hbar}} e^{i\hbar\frac{\partial_p}{2}(\partial_x-\partial_{x'})} \vec{\psi}^\dagger(x+y;\mathfrak{a})\hat{h}_{x''+y''}(e^{i\mathfrak{a}p/\hbar})\vec{\psi}(x'+y';\mathfrak{a})\Big|_{\substack{x''=x'=x \\ y''=y'=y}} . \quad (94)
$$

We choose the coupling constants in such a way that, in the limit $\mathfrak{a} \to 0$, the symbol of the Hamiltonian approaches infinity for all momenta but a finite number of them. We then expand the symbol around such points. Physically, the infinities correspond to infinite excitation

energies, and the momenta at which the symbol is not divergent describe the lowest (finite) excitations. By definition, the symbol is a $2\pi$-periodic function of $\frac{\mathfrak{a}p}{\hbar}$, therefore the expansion in the momentum is also an expansion in the lattice spacing $\mathfrak{a}$. The divergency of some coupling constants will compensate the presence of powers of $\mathfrak{a}$ in the terms of the expansion. Since generally we consider Hamiltonians with a finite number of coupling constants, only a finite number of the terms of the expansion will remain nonzero in the limit $\mathfrak{a} \to 0$. In other words, in the scaling limit the symbol becomes a polynomial in the momentum. This is sufficient for the QFT Hamiltonian to be local (in (94), each power of $p$ becomes a derivative with respect to $(x-y)$).

As a prototypical example, we consider the quantum XY model, whose Hamiltonian's symbol is given by

$$\hat{h}_x(e^{i\mathfrak{a}p/\hbar}) = 2J_x(\mathfrak{a})(h_x(\mathfrak{a}) - \cos(\mathfrak{a}p/\hbar))\sigma^y - 2J_x(\mathfrak{a})\gamma_x(\mathfrak{a})\sin(\mathfrak{a}p/\hbar)\sigma^x. \tag{95}$$

Let us determine in which scaling limits the symbol remains finite for $\mathfrak{a}p \approx 0$. We have

$$\hat{h}_x(e^{i\mathfrak{a}p/\hbar}) \approx 2J_x(\mathfrak{a})\left(h_x(\mathfrak{a}) - 1 + \frac{\mathfrak{a}^2p^2}{2\hbar^2} + O\left(\mathfrak{a}^4p^4/\hbar^4\right)\right)\sigma^y - 2J_x(\mathfrak{a})\gamma_x(\mathfrak{a})\left(\frac{\mathfrak{a}p}{\hbar} + O\left(\mathfrak{a}^3p^3/\hbar^3\right)\right)\sigma^x. \tag{96}$$

Essentially, there are three options for simplifying the divergencies:

$$\begin{bmatrix} h_x(\mathfrak{a}) \approx 1 - \dfrac{\gamma_x m_x c_x \mathfrak{a}}{\hbar} \\ 2J_x(\mathfrak{a})\gamma_x(\mathfrak{a}) \approx -\hbar\dfrac{c_x}{\mathfrak{a}} \\ \gamma_x(\mathfrak{a}) = \gamma_x \end{bmatrix} \qquad m_x c_x^2 \sigma^y + p c_x \sigma^x \tag{97a}$$

$$\begin{bmatrix} h_x(\mathfrak{a}) \approx 1 - \dfrac{m_x c_x^2 \mu_x \mathfrak{a}^2}{\hbar^2} \\ J_x(\mathfrak{a}) \approx -\dfrac{\hbar^2}{2\mu_x \mathfrak{a}^2} \\ \gamma_x(\mathfrak{a}) \approx \dfrac{\mu_x c_x \mathfrak{a}}{\hbar} \end{bmatrix} \qquad \left(m_x c_x^2 - \dfrac{p^2}{2\mu_x}\right)\sigma^y - p c_x \sigma^x \tag{97b}$$

$$\begin{bmatrix} 2J_x(\mathfrak{a})\gamma_x(\mathfrak{a}) = \hbar\dfrac{c_x}{\mathfrak{a}} \\ 2J_x(\mathfrak{a})(h_x(\mathfrak{a}) - 1) = m_x c_x^2 \\ h_x(\mathfrak{a}) = h_x \end{bmatrix} \qquad c_x(m_x c_x \sigma^y - p\sigma^x) \oplus c_x\left(\dfrac{h_x + 1}{h_x - 1}m_x c_x \sigma^y + p\sigma^x\right), \tag{97c}$$

where, on the right hand side, we showed the corresponding QFT symbols. Case (97a) is the Ising QFT (see, *e.g.*, [101]); case (97b) describes the multicritical point $\gamma = 0, h = 1$; case (97c) describes the limit of infinite anisotropy. In case (97c), the symbol is the direct sum of two terms because, in the scaling limit, there is also another momentum at which the symbol is finite, namely, $\frac{\mathfrak{a}p}{\hbar} = \pi$; the last term describes that contribution (with the momentum shifted by $\pi$). We note that, in (97c), in order to write the corresponding Hamiltonian, one should redefine the Majorana field in such a way to distinguish even and odd sites.

In the following, we will restrict to case (97a) with homogeneous couplings, namely

$$\lim_{\mathfrak{a}\to 0} H = \frac{1}{4}\int_{-\infty}^{\infty} dx\, \vec{\psi}(x) \cdot [mc^2\sigma^y + ic\partial_x\sigma^x]\vec{\psi}(x). \tag{98}$$

Time evolution from a locally quasi-stationary state is then described by the Moyal equation

$$i\hbar\partial_t \rho_{x,t}(p) = \varepsilon^{QFT}(p) \star \rho_{x,t}(p) - \rho_{x,t}(p) \star \varepsilon^{QFT}(p), \tag{99}$$

with $\varepsilon^{QFT} = c\sqrt{m^2c^2 + p^2}$. We aim at comparing this result with the continuum limit of the lattice time evolution. We assume the initial state to be an LQSS. The lattice dynamics are solved by (45); there, the Hamiltonian enters through the quantity

$$\omega^{\text{lat}}(p,q;\mathfrak{a}) = \varepsilon\left(p + \frac{q}{2}\right) - \varepsilon\left(p - \frac{q}{2}\right) \qquad p \in \left(-\frac{\pi}{\mathfrak{a}}, \frac{\pi}{\mathfrak{a}}\right), \quad q \in \mathbb{R}, \tag{100}$$

where we highlighted the dependence on the lattice spacing. The QFT is the result of the expansion about the zeros of the dispersion relation, but (100) has additional zeros. In our specific case, we can immediately identify $\frac{\mathfrak{a}}{\hbar}q = 0$, $\frac{\mathfrak{a}}{\hbar}q = 2\pi$, and $\frac{\mathfrak{a}}{\hbar}p = \pi$, but there can also be further zeros. For example, for $\gamma > \sqrt{2}$ there are closed curves centred at $(\frac{\mathfrak{a}}{\hbar}p, \frac{\mathfrak{a}}{\hbar}q) = (\pi, 0)$ and $(\frac{\mathfrak{a}}{\hbar}p, \frac{\mathfrak{a}}{\hbar}q) = (0, 2\pi)$ with $\omega^{\text{lat}}(p,q;\mathfrak{a}) = 0$. For the sake of simplicity we assume $\gamma = 1$ (quantum Ising model), for which we find (if not stated otherwise, $\frac{\mathfrak{a}q}{2\hbar} \in (-\pi, \pi)$ and $\frac{\mathfrak{a}p}{\hbar} \in (-\pi, \pi)$)

$$\lim_{\mathfrak{a} \to 0} \omega^{\text{lat}}(p,q;\mathfrak{a}) =$$

$$\begin{cases}
-cq\,\text{sgn}(p)\cos\frac{\mathfrak{a}p}{2\hbar} & \frac{\mathfrak{a}q}{2\hbar} \approx 0 & \frac{\mathfrak{a}p}{\hbar} \not\approx 0, \pi \\
-2cp\,\text{sgn}(q)\cos\frac{\mathfrak{a}q}{4\hbar} & \frac{\mathfrak{a}p}{\hbar} \approx 0 & \frac{\mathfrak{a}q}{2\hbar} \not\approx 0, \pi \\
c(q - \frac{2\pi\hbar}{\mathfrak{a}})\sin\frac{\mathfrak{a}p}{2\hbar} & \frac{\mathfrak{a}q}{2\hbar} \approx \pi & \frac{\mathfrak{a}p}{\hbar} \not\approx 0, \pi & \frac{\mathfrak{a}q}{2\hbar} \in (0, 2\pi) \\
2c(p - \frac{\pi\hbar}{\mathfrak{a}})\sin\frac{\mathfrak{a}q}{4\hbar} & \frac{\mathfrak{a}p}{\hbar} \approx \pi & \frac{\mathfrak{a}q}{2\hbar} \not\approx 0, \pi & \frac{\mathfrak{a}p}{\hbar} \in (0, 2\pi) \\
c\sqrt{m^2c^2 + (p - q/2)^2} - c\sqrt{m^2c^2 + (p + q/2)^2} & \frac{\mathfrak{a}p}{\hbar}, \frac{\mathfrak{a}q}{2\hbar} \approx 0 \\
c\sqrt{m^2c^2 + (p - q/2)^2} - c\sqrt{m^2c^2 + (p + q/2)^2} & \frac{\mathfrak{a}p}{\hbar} \approx \pi & \frac{\mathfrak{a}q}{2\hbar} \approx \pi & \frac{\mathfrak{a}p}{\hbar}, \frac{\mathfrak{a}q}{2\hbar} \in (0, 2\pi) \\
0 & \frac{\mathfrak{a}p}{\hbar} \approx 0 & \frac{\mathfrak{a}q}{2\hbar} \approx \pi & \frac{\mathfrak{a}q}{2\hbar} \in (0, 2\pi) \\
0 & \frac{\mathfrak{a}p}{\hbar} \approx \pi & \frac{\mathfrak{a}q}{2\hbar} \approx 0 & \frac{\mathfrak{a}p}{\hbar} \in (0, 2\pi).
\end{cases} \tag{101}$$

So far we have only considered the limit $\mathfrak{a} \to 0$ in the terms dependent on the Hamiltonian. It is however clear that, if we aim at capturing the continuum scaling limit by the QFT describing the low-energy physics, also the initial state should be described by the QFT. Specifically, excitations that, in the limit $\mathfrak{a} \to 0$, have infinite energy, should not be present in the state, that is to say

$$\lim_{\mathfrak{a} \to 0} \rho_{x,0}(p) = 0 \qquad \frac{\mathfrak{a}p}{\hbar} \not\approx 0. \tag{102}$$

In addition, every nontrivial root density in the lattice for which $\exists \lim_{x \to \pm\infty} \rho_x(p) = \rho_{\pm}(p)$ will approach a step function in the limit $\mathfrak{a} \to 0$; let us then assume straight away that $\rho_{x,0}(t)$ is like in (80). This allows us to simplify the lattice solution in the limit $\mathfrak{a} \to 0$

$$\rho_{x,t}(p) \sim \rho_-(p) + [\rho_+(p) - \rho_-(p)] \lim_{\tilde{\epsilon} \to 0} \lim_{\mathfrak{a} \to 0} \lim_{\epsilon \to 0} \left[ \int_{-\frac{\tilde{\epsilon}\hbar}{\mathfrak{a}}}^{\frac{\tilde{\epsilon}\hbar}{\mathfrak{a}}} \frac{\mathrm{d}q}{4\pi} e^{\frac{2iqx}{\hbar}} e^{-ict\frac{\sqrt{m^2c^2+(p-q)^2} - \sqrt{m^2c^2+(p+q)^2}}{\hbar}} \frac{1}{iq + \epsilon} + \right.$$

$$\left. \int_{\pi > |q| > \tilde{\epsilon}} \frac{\mathrm{d}k}{4\pi} e^{\frac{2ikx}{\mathfrak{a}}} e^{it\frac{2cp\,\text{sgn}(k)\cos\frac{k}{2}}{\hbar}} \left[\text{sgn}(0) - i\cot\frac{k - i\epsilon}{2}\right] \right] =$$

$$\rho_-(p) + [\rho_+(p) - \rho_-(p)] \lim_{\epsilon \to 0} \int_{-\infty}^{\infty} \frac{\mathrm{d}q}{4\pi} e^{\frac{2iqx}{\hbar}} e^{-ict\frac{\sqrt{m^2c^2+(p-q)^2} - \sqrt{m^2c^2+(p+q)^2}}{\hbar}} \frac{1}{iq + \epsilon}, \tag{103}$$

valid for $\frac{\mathfrak{a}p}{\hbar} \approx 0$, otherwise $\rho_{x,t}(p) = 0$. As expected, the final result is the time evolution in the QFT, which follows equation (99).

Importantly, in order to recover the QFT result, we had to assume the initial state to be describable by the QFT. In other words, as testified by (101), in the continuum scaling limit time evolution is not sufficient to extinguish highly energetic degrees of freedom (*i.e.*, in the quantum Ising model, excitations with $\frac{\mathfrak{a}p}{\hbar} \not\approx 0$).

# 6 Conclusion

In this paper we have studied the fundamentals of generalised hydrodynamics in noninteracting spin chains. GHD was originally developed in a perturbative framework, as the asymptotic solution of time evolution in the limit of large time [35,36] or low inhomogeneity [56,57] in integrable systems. Ref. [69] made a first attempt to lift the theory into a non-perturbative level by conjecturing the existence of so-called "locally quasi-stationary states", which are states completely characterised by a local version of the root densities of the thermodynamic Bethe Ansatz [87]. This suggestion was however not exploited, and the main efforts were rather put in determining the next orders of perturbation theory. There have been indeed proposals to go beyond generalised hydrodynamics, even in the presence of interactions [79,83]. The uncertainty of what should be captured by generalised hydrodynamics has however made it problematic even the determination of the next orders of perturbation theory in noninteracting spin chains without $U(1)$ symmetry [84].

In order to resolve this issue in noninteracting spin chains, we have adopted here the alternative perspective of identifying the invariant subspace made of the locally quasi-stationary states. In one-site shift invariant models with $U(1)$ symmetry, like the XX model, this is a straightforward, intuitive step; this paper provides a solution in the more complicated situation when the number of Jordan-Wigner fermions is not conserved. Once the subspace of locally quasi-stationary states is identified, *GHD becomes the theory describing time evolution exactly, indeed the time evolving state is completely determined by the continuity equations satisfied by the charges*. This observation has allowed for a non-perturbative derivation of generalised hydrodynamics (*cf*. Section 3.2.3).

We would like to emphasise that, even if first-order GHD becomes exact only at the Euler scale, the theory does not assume the absence of correlations. Generalised hydrodynamics is a mapping of the initial state into the state at nonzero time; if there are quantum correlations in the initial state, they will be "transported" by GHD. At the first order, the mapping does not introduce new quantum correlations, but the state will still be quantum correlated, even at a finite time. Ref. [74] provides an explicit example of this phenomenon, as it proposes a framework to compute quantum correlations within first-order GHD.

We have also found other invariant subspaces that "cut" the Hilbert space in a different way - Section 3.5. While the existence of so many invariant subspaces could seem extraordinary, Ref. [102] has actually shown that time evolution occurs in a tiny part of the Hilbert space even in generic systems; we then wonder whether approximately invariant subspaces could be defined also in non-integrable models.

**Interacting integrable systems.** Our findings are restricted to noninteracting spin chains, but we think that a similar change of perspective could be useful also in the presence of interactions. Specifically, our calculations are based on the possibility to define a complete set o quasi-local charge densities in such a way to make all their currents conserved. Such charge densities are then the building blocks for the invariant subspace of locally quasi-stationary states. This very preliminary step is already questionable in interacting integrable systems. For example, in the gapped XXZ spin-$\frac{1}{2}$ chain, there is evidence that there is a single (quasi-)local charge that is odd under spin flip: the total spin in the direction of the anisotropy. This is enough to exclude the existence of a subspace consisting of LQSSs with a nontrivial profile of the sign of the magnetisation. The results of Ref. [54] go in this direction, indeed the authors had to introduce an auxiliary parameter to describe the time evolution of states without a fixed sign of the local magnetisation. Nevertheless, we do not exclude, and, in fact, we believe, that even in the gapped XXZ model there are two main invariant subspaces consisting of LQSSs with a fixed sign of the local magnetisation. Such invariant subspaces could be

investigated within Bethe anstaz, quite independently of the knowledge of the equations of motion describing time evolution within the subspaces, which is instead the program of GHD. As a matter of fact, candidates for LQSSs in interacting integrable systems already exist: for example, they could be stationary states of inhomogeneous integrable models (obtained by introducing inhomogeneities in the spectral parameters of the R-matrix - see, *e.g.*, Ref. [103]). In addition, it is worth observing that, assuming all the quantities used to describe the system to be defined in the appropriate way, all the equations that we obtained could have been guessed by recognising that products in the homogenous case become Moyal star products in the inhomogeneous one. This simple rule could be specific to noninteracting systems, but it is still reasonable to expect that also in interacting integrable models the TBA equations should be modified in some analogous way. More specifically, in the light of our identification of the root density with a Wigner function, the generalisation to the interacting case can be seen as the conception of a phase-space representation of the Bethe Ansatz structure.

Finally, we have observed that the subspace of LQSSs is not only invariant under the effect of a homogeneous Hamiltonian, but also if the Hamiltonian is replaced by a locally quasi-conserved operator (*cf.* (67)): the algebra of the locally quasi-conserved operators is closed. We wonder whether this property holds true also in the presence of interactions.

## Acknowledgements

I thank Bruno Bertini for discussions and Andrei Zvyagin for constructive comments.

**Funding information.** This work was supported by a grant LabEx PALM (ANR-10-LABX-0039-PALM) and by the European Research Council under the Starting Grant No. 805252 LoCoMacro.

## A  Non-equilibrium dynamics in inhomogeneous systems

In order to ease the notations, in this appendix the lattice spacing $\mathfrak{a}$ and the reduced Planck constant $\hbar$ are set to unity.

In noninteracting models, the correlation matrix (19) satisfies a reduced version of the von Neumann equation (*cf.* (30))

$$\partial_t \Gamma(t) + i[\mathcal{H}, \Gamma(t)]_- = 0, \tag{104}$$

where $\mathcal{H}$ was defined in (13). In homogeneous systems, this can be reduced further to an analogous equation satisfied by the symbols $\hat{\Gamma}(e^{ik})$ and $\hat{h}(e^{ik})$ (the homogeneous version of (14) and (20))

$$\partial_t \hat{\Gamma}_t(z_p) + i\big[\hat{h}(z_p), \hat{\Gamma}_t(z_p)\big]_- = 0. \tag{105}$$

This equation is extremely powerful: the time evolution of a quantum many-body system is recast into the time evolution of a $(2\kappa)$-by-$(2\kappa)$ matrix, which depends on a parameter, the momentum $p$, playing the mere role of a spectator.

In this section, we show how to generalise this result to inhomogeneous systems. To that aim, we parametrise the correlation matrix and the Hamiltonian as follows

$$\Gamma_{ij}^{\ell n}(t) = \left[\Gamma_{\frac{\ell+n}{2},t}^{(n-\ell)}\right]_{ij} \qquad \mathcal{H}_{ij}^{\ell n} = \left[\mathcal{H}_{\frac{\ell+n}{2}}^{(n-\ell)}\right]_{ij}. \tag{106}$$

By writing the indices explicitly, (104) reads as

$$\partial_t \Gamma_{x,t}^{(2r)} + i \sum_{j\in\mathbb{Z}} \mathcal{H}_{x+\frac{j}{2}}^{(2r+j)} \Gamma_{x+r+\frac{j}{2},t}^{(-j)} - \Gamma_{x+\frac{j}{2},t}^{(2r+j)} \mathcal{H}_{x+r+\frac{j}{2}}^{(-j)} = 0 \qquad x+r \in \mathbb{Z}. \tag{107}$$

This can be rewritten as follows

$$\int_{-\pi}^{\pi} \frac{dp}{2\pi} e^{-2irp} \partial_t \hat{\Gamma}_{x,t}^{\text{phys}}(z_p) +$$

$$i \sum_{j\in\mathbb{Z}} \iint_{-\pi}^{\pi} \frac{dpdq}{(2\pi)^2} e^{-i(2r+j)p} e^{ijq} \hat{h}_{x+\frac{j}{2}}^{\text{phys}}(z_p) \hat{\Gamma}_{x+r+\frac{j}{2},t}^{\text{phys}}(z_q) -$$

$$i \sum_{j\in\mathbb{Z}} \iint_{-\pi}^{\pi} \frac{dpdq}{(2\pi)^2} e^{-i(2r+j)p} e^{ijq} \hat{\Gamma}_{x+\frac{j}{2},t}^{\text{phys}}(z_p) \hat{h}_{x+r+\frac{j}{2}}^{\text{phys}}(z_q) = 0, \tag{108}$$

where the physical symbols are defined as

$$\hat{h}_x^{\text{phys}}(z_p) = \sum_{j\in x+\mathbb{Z}} z_p^{2j} \mathcal{H}_x^{(2j)} \qquad \hat{\Gamma}_{x,t}^{\text{phys}}(z_p) = \sum_{j\in x+\mathbb{Z}} z_p^{2j} \Gamma_{x,t}^{(2j)}. \tag{109}$$

We note that we can replace the physical symbols by the corresponding symbols (16) extended to $x \in \mathbb{R}$. Since the first line of (108) is trivial and the third line is analogous to the second one, we focus on the second line. We have

$$i \sum_{j\in\mathbb{Z}} \iint_{-\pi}^{\pi} \frac{dpdq}{(2\pi)^2} e^{-i(2r+j)p} e^{ijq} \hat{h}_{x+\frac{j}{2}}^{\text{phys}}(z_p) \hat{\Gamma}_{x+r+\frac{j}{2},t}^{\text{phys}}(z_q) =$$

$$i \sum_{j\in\mathbb{Z}} \iint_{-\pi}^{\pi} \frac{dpdq}{(2\pi)^2} e^{-2i(r+j)p} e^{2ijq} \hat{h}_{x+j}^{\mathfrak{p}}(z_p) \hat{\Gamma}_{x+r+j,t}^{+}(z_q) +$$

$$i \sum_{j\in\mathbb{Z}} \iint_{-\pi}^{\pi} \frac{dpdq}{(2\pi)^2} e^{-2i(r+j)p} e^{2ijq} e^{i(p-q)} \hat{h}_{x+\frac{2j-1}{2}}^{-\mathfrak{p}}(z_p) \hat{\Gamma}_{x+r+\frac{2j-1}{2},t}^{-}(z_q), \tag{110}$$

where we split the sum in two parts and defined $\mathfrak{p} = (-1)^{2x}$. We can get rid of the index $j$ in the subscripts by means of identities like

$$\hat{h}_{x+j}^{\pm}(z_p) = e^{j\partial_x} \hat{h}_x^{\pm}(z_p), \tag{111}$$

which are exact if the extensions are entire functions of $x$, which we are assuming. We can therefore rewrite the expression as follows

$$(110) = i \sum_{j\in\mathbb{Z}} \iint_{-\pi}^{\pi} \frac{dpdq}{(2\pi)^2} e^{-2i(r+j)p} e^{2ijq} e^{j\partial_x} e^{(r+j)\partial_y} \hat{h}_x^{\mathfrak{p}}(z_p) \hat{\Gamma}_{y,t}^{+}(z_q) \Big|_{y=x} +$$

$$i \sum_{j\in\mathbb{Z}} \iint_{-\pi}^{\pi} \frac{dpdq}{(2\pi)^2} e^{-2i(r+j)p} e^{2ijq} e^{i(p-q)} e^{\frac{2j-1}{2}\mathfrak{a}\partial_x} e^{(r+\frac{2j-1}{2})\partial_y} \hat{h}_x^{-\mathfrak{p}}(z_p) \hat{\Gamma}_{y,t}^{-}(z_q) \Big|_{y=x}. \tag{112}$$

By repeated integrations by parts, the coefficients of the space derivatives can be rewritten as derivatives with respect to the momentum

$$(110) = i \sum_{j\in\mathbb{Z}} \iint_{-\pi}^{\pi} \frac{dpdq}{(2\pi)^2} e^{-2i(r+j)p} e^{2ijq} e^{i\frac{\partial_q\partial_x - \partial_p\partial_y}{2}} \hat{h}_x^{\mathfrak{p}}(z_p) \hat{\Gamma}_{y,t}^{+}(z_q) \Big|_{y=x} +$$

$$i \sum_{j\in\mathbb{Z}} \iint_{-\pi}^{\pi} \frac{dpdq}{(2\pi)^2} e^{-2i(r+j)p} e^{2ijq} e^{i(p-q)} e^{i\frac{\partial_q\partial_x - \partial_p\partial_y}{2}} \hat{h}_x^{-\mathfrak{p}}(z_p) \hat{\Gamma}_{y,t}^{-}(z_q) \Big|_{y=x}. \tag{113}$$

The sum over $j$ forces $q$ to be either equal to $p$ or to $p + \pi$; using the transformation properties of the symbols under a shift of the momentum by $\pi$, (15), we then find

$$(110) = i \sum_{j \in \mathbb{Z}} \int_{-\pi}^{\pi} \frac{dp}{2\pi} e^{-2irp} e^{i \frac{\partial_q \partial_x - \partial_p \partial_y}{2}} \hat{h}_x^{\mathfrak{p}}(z_p) \hat{\Gamma}_{y,t}^+(z_q) \Big|_{\substack{q=p \\ y=x}} +$$

$$i \sum_{j \in \mathbb{Z}} \int_{-\pi}^{\pi} \frac{dp}{2\pi} e^{-2irp} e^{i \frac{\partial_q \partial_x - \partial_p \partial_y}{2}} \hat{h}_x^{-\mathfrak{p}}(z_p) \hat{\Gamma}_{y,t}^-(z_q) \Big|_{\substack{q=p \\ y=x}} . \quad (114)$$

Putting all the pieces together we get (*cf.* (108))

$$\partial_t \hat{\Gamma}_{x,t}^{\mathfrak{p}}(z_p) + i e^{i \frac{\partial_q \partial_x - \partial_p \partial_y}{2}} \left( \hat{h}_x^{\mathfrak{p}}(z_p) \hat{\Gamma}_{y,t}^+(z_q) + \hat{h}_x^{-\mathfrak{p}}(z_p) \hat{\Gamma}_{y,t}^-(z_q) - \right.$$
$$\left. \hat{\Gamma}_{x,t}^{\mathfrak{p}}(z_p) \hat{h}_y^+(z_q) - \hat{\Gamma}_{x,t}^{-\mathfrak{p}}(z_p) \hat{h}_y^-(z_q) \right) \Big|_{\substack{q=p \\ y=x}} = 0 \qquad \mathfrak{p} = (-1)^{2x} . \quad (115)$$

This equation describes only the time evolution of the physical part of the correlation matrix. Indeed, strictly speaking, it holds only at the values of $x$ such that $\mathfrak{p} = (-1)^{2x}$. We fix the time evolution of the unphysical part by asking for this equation to hold at any position. Since, in this way, $x$ and $\mathfrak{p}$ become independent, we can sum the two equations corresponding to $\mathfrak{p} = \pm 1$, obtaining

$$\partial_t \hat{\Gamma}_{x,t}(z_p) + i e^{i \frac{\partial_q \partial_x - \partial_p \partial_y}{2}} \left( \hat{h}_x(z_p) \hat{\Gamma}_{y,t}(z_q) - \hat{\Gamma}_{x,t}(z_p) \hat{h}_y(z_q) \right) \Big|_{\substack{q=p \\ y=x}} = 0 . \quad (116)$$

This is the most important result of this section, being the generalization of (105) to the inhomogeneous case. It's a Wigner description of the dynamics, and is valid also in noninteracting systems without a $U(1)$ symmetry (we remind the reader that the standard setting where the so-called Wigner function comes into play is the time evolution of a noninteracting state with a fixed number of particles). The symbol of the correlation matrix could be interpreted as a Wigner function with an internal degree of freedom, but we prefer to identify the Wigner function(s) with the localized version of the root density(ies), as will be discussed in the next section.

## B  Weak inhomogeneous limit: an asymptotic expansion

In order to ease the notations, in this appendix the lattice spacing $\mathfrak{a}$ and the reduced Planck constant $\hbar$ are set to unity.

   If time evolution is generated by a time independent noninteracting Hamiltonian, (116) is solved by (36), which, in an XY-like model with $\phi(p) = 0$, can be written as

$$\hat{\Gamma}_{x,t}(z_p) = \sum_{s,s'=\pm 1} \iint \frac{dy\, dq}{2\pi} e^{-i\left\{ q(y-x) + t\left[ s'\varepsilon\left(s'p + s'\frac{q}{2}\right) - s\varepsilon\left(sp - s\frac{q}{2}\right) \right] \right\}} \times$$

$$\frac{1 + s'\sigma^y e^{i\vec{\theta}\left(p + \frac{q}{2}\right) \cdot \vec{\sigma}}}{2} \hat{\Gamma}_{y,0}(z_p) \frac{1 + s\sigma^y e^{i\vec{\theta}\left(p - \frac{q}{2}\right) \cdot \vec{\sigma}}}{2} . \quad (117)$$

In this appendix we work out an asymptotic expansion of this equation in the limit of low inhomogeneity. Let $\xi$ be the typical length scale of the inhomogeneity in the initial state. We consider the class of initial states described by an analogous symbol with a different typical

length scale $\chi\xi$. Their symbols time evolve as follows

$$\hat{\Gamma}^{[\chi]}_{\chi x, \chi t}(z_p) = \chi \sum_{s,s' \pm 1} \iint \frac{\mathrm{d}y\,\mathrm{d}q}{2\pi} e^{-i\chi\left\{q(y-x)+t[s'\varepsilon(s'p+s'\frac{q}{2})-s\varepsilon(sp-s\frac{q}{2})]\right\}} \times$$
$$\frac{1+s'\sigma^y e^{i\vec{\theta}(p+\frac{q}{2})\cdot\vec{\sigma}}}{2} \hat{\Gamma}_{y,0}(z_p) \frac{1+s\sigma^y e^{i\vec{\theta}(p-\frac{q}{2})\cdot\vec{\sigma}}}{2}, \quad (118)$$

where we rescaled the integration variable $y$, the distance $x$, and the time $t$. Introducing the integration variable

$$\phi_{s,s'}(y,q) = \chi \left[ y - x + t \frac{s'\varepsilon\left(s'p+s'\frac{q}{2}\right)-s\varepsilon\left(sp-s\frac{q}{2}\right)-s'\varepsilon(s'p)+s\varepsilon(sp)-q\frac{v(s'p)+v(sp)}{2}}{q} \right], \quad (119)$$

the integral reads as

$$\hat{\Gamma}^{[\chi]}_{\chi x, \chi t}(z_p) = \sum_{s,s'} e^{-i\chi t(s'\varepsilon(s'p)+s\varepsilon(-sp))} \int \mathrm{d}q \frac{1+s'\sigma^y e^{i\vec{\theta}(p+\frac{q}{2})\cdot\vec{\sigma}}}{2} \times$$
$$\int \frac{\mathrm{d}\phi}{2\pi} e^{-i\phi q} \hat{\Gamma}_{x+\frac{\phi}{\xi}-t\frac{s'\varepsilon\left(s'p+s'\frac{q}{2}\right)-s\varepsilon\left(sp-s\frac{q}{2}\right)-s'\varepsilon(s'p)+s\varepsilon(sp)}{q},0}(z_p) \frac{1+s\sigma^y e^{i\vec{\theta}(p-\frac{q}{2})\cdot\vec{\sigma}}}{2}. \quad (120)$$

We are interested in the asymptotic expansion in the limit of large $\chi$. Asymptotically we have

$$\hat{\Gamma}^{[\chi]}_{\chi x, \chi t}(z_p) \sim \sum_{s,s'} e^{-i\chi t(s'\varepsilon(s'p)-s\varepsilon(sp))} \int \mathrm{d}q \frac{1+s'\sigma^y e^{i\vec{\theta}(p+\frac{q}{2})\cdot\vec{\sigma}}}{2} \times$$
$$\int \frac{\mathrm{d}\phi}{2\pi} e^{-i\phi q} \sum_{n=0}^{\infty} \frac{\phi^n}{\chi^n n!} \partial_x^n \hat{\Gamma}_{x-t\frac{s'\varepsilon(s'p+s'q/2)-s\varepsilon\left(sp-s\frac{q}{2}\right)-s'\varepsilon(s'p)+s\varepsilon(sp)}{q},0}(z_p) \frac{1+s\sigma^y e^{i\vec{\theta}(p-\frac{q}{2})\cdot\vec{\sigma}}}{2}. \quad (121)$$

Powers of $\phi$ can be seen as derivatives with respect to $-iq$ of $e^{-i\phi q}$, so repeating integration by parts gives

$$\hat{\Gamma}_{x,t}(z_p) \sim \sum_{n=0}^{\infty} \frac{(-i)^n}{n!} \sum_{s,s'=\pm 1} e^{-it[s'\varepsilon(s'p)-s\varepsilon(sp)]} \partial_q^n \left\{ \frac{1+s'\sigma^y e^{i\vec{\theta}(p+\frac{q}{2})\cdot\vec{\sigma}}}{2} \times \right.$$
$$\left. \partial_x^n \hat{\Gamma}_{x-t\frac{s'\varepsilon\left(s'p+s'\frac{q}{2}\right)-s\varepsilon\left(sp-s\frac{q}{2}\right)-s'\varepsilon(s'p)+s\varepsilon(sp)}{q},0}(z_p) \frac{1+s\sigma^y e^{i\vec{\theta}(p-\frac{q}{2})\cdot\vec{\sigma}}}{2} \right\}_{q=0}. \quad (122)$$

Here we rescaled position and time back to origin, so that $\chi$ has disappeared from the equation. We stress however that the $n$-th term of the sum is suppressed by a factor $\chi$ with respect to the $(n-1)$-th term under a rescaling of space and time by $\chi$; the truncation of the series at $n = N$ approximates the correlation matrix up to $O\left(1/\chi^{N+1}\right)$ corrections.

We identify two different contributions:

$s' = s$: time evolution causes only the mixing of information from different parts of the chain; we will refer to this as the *hydrodynamic* part of time evolution;

$s' \neq s$: time evolution is characterised by rapidly oscillatory terms; we will refer to this as the *purely quantum* part of time evolution.

**Definition.** *A locally quasi-stationary state is a state with a null purely quantum part.*

**Definition.** *An off-diagonal state is a state with a null hydrodynamic part.*

Since in this appendix we only aim at a perturbative solution to the dynamics, we consider the following expansion:

$$\hat{\Gamma}_{x,t}(z) = \sum_{j=0}^{\infty} \partial_x^j \Gamma_{x,t}^{(j)}(z). \tag{123}$$

Imposing (122) order by order gives the infinite system of equations

$$\hat{\Gamma}_{x,t}^{(m)}(e^{ip}) = \sum_{n=0}^{m} \frac{(-i)^n}{n!} \sum_{s,s'=\pm 1} e^{-it[s'\varepsilon(s'p)-s\varepsilon(sp)]} \partial_q^n \left\{ \frac{1+s'\sigma^y e^{i\vec{\theta}\left(p+\frac{q}{2}\right)\cdot\vec{\sigma}}}{2} \times \right.$$
$$\left. \hat{\Gamma}^{(m-n)}_{x-t\frac{s'\varepsilon\left(s'p+s'\frac{q}{2}\right)-s\varepsilon\left(sp-s\frac{q}{2}\right)-s'\varepsilon(s'p)+s\varepsilon(sp)}{q},0} (e^{ip}) \frac{1+s\sigma^y e^{i\vec{\theta}\left(p-\frac{q}{2}\right)\cdot\vec{\sigma}}}{2} \right\}_{q=0}. \tag{124}$$

### B.1 Locally quasi-stationary states

A locally quasi-stationary state is completely characterised by the equations

$$\sum_{n=0}^{m} \frac{(-i)^n}{n!} \partial_q^n \left\{ \frac{1-s\sigma^y e^{i\vec{\theta}\left(p+\frac{q}{2}\right)\cdot\vec{\sigma}}}{2} \times \right.$$
$$\left. \hat{\Gamma}^{(m-n)}_{x-st\frac{\varepsilon(-sp)+\varepsilon(sp)-\varepsilon\left(-sp-s\frac{q}{2}\right)-\varepsilon\left(sp-s\frac{q}{2}\right)}{q}} (e^{ip}) \frac{1+s\sigma^y e^{i\vec{\theta}\left(p-\frac{q}{2}\right)\cdot\vec{\sigma}}}{2} \right\}_{q=0} = 0 \qquad s=\pm 1. \tag{125}$$

By truncating the number of equations to $m = N$, the error is $O\left(1/\chi^{N+1}\right)$. A single function $\rho_x(p)$, which we identify with the *root density*, completely characterises a locally quasi-stationary state; the first terms of the expansion of its symbol are the following

$$\hat{\Gamma}_x^{(0)}(e^{ip}) = \sum_s [4\pi\rho_x(sp)-1]s\frac{I+s\sigma^y e^{i\theta(p)\sigma^z}}{2}$$

$$\hat{\Gamma}_x^{(1)}(e^{ip}) = -\pi\theta'(p)\sigma^z \sum_s s\rho_x(sp)$$

$$\Gamma_x^{(2)}(e^{ip}) = \pi\frac{\theta''(p)}{4}\sigma^x e^{i\theta(p)\sigma^z} \sum_s \rho_x(sp)$$

$$\Gamma_x^{(3)}(e^{ip}) = \pi\frac{\theta'''(p)+2\theta'(p)^3}{24}\sigma^z \sum_s s\rho_x(sp) \tag{126}$$

$$\Gamma_x^{(4)}(e^{ip}) = -\pi\frac{\theta^{\mathrm{iv}}(p)}{192}\sigma^x e^{i\theta(p)\sigma^z} \sum_s \rho_x(sp)$$

$$\Gamma_x^{(5)}(e^{ip}) = -\pi\frac{16\theta'(p)^5+20\theta'(p)^2\theta'''(p)+\theta^{\mathrm{v}}(p)}{1920}\sigma^z \sum_s s\rho_x(sp).$$

We have defined $\rho_x(p)$ in such a way that, at the leading order, it corresponds to the spuriously local root density $\rho_x^{\mathrm{fake}}(p)$ as defined in (21).

**Time evolution.** The symbol of a locally quasi-stationary state time evolves as follows:

$$\partial_x^m \hat{\Gamma}_{x,t}^{(m)}(e^{ip}) = \sum_{n=0}^m \frac{(-i)^n}{n!} \sum_{s=\pm 1} \partial_q^n \left\{ \frac{1+s\sigma^y e^{i\vec{\theta}\left(p+\frac{q}{2}\right)\cdot\vec{\sigma}}}{2} \times \right.$$
$$\left. \partial_x^m \hat{\Gamma}_{x-st\frac{\varepsilon\left(sp+s\frac{q}{2}\right)-\varepsilon\left(sp-s\frac{q}{2}\right)}{q},0}^{(m-n)}(e^{ip}) \frac{1+s\sigma^y e^{i\vec{\theta}\left(p-\frac{q}{2}\right)\cdot\vec{\sigma}}}{2} \right\}_{q=0}, \quad (127)$$

where $\hat{\Gamma}_{x,t}^{(n)}(e^{ip})$ are written as in (126). Up to $O\left(1/\chi^5\right)$ corrections, the root density time evolves as follows:

$$\rho_{x,t}(p) \approx \left(1 + \frac{t}{24}v''(p)\partial_x^3 - \frac{t}{1920}v^{iv}(p)\partial_x^5 + \frac{t^2}{1152}v''(p)^2\partial_x^6\right)$$
$$\left(\rho_{x-v(p)t,0}(p) + \frac{3\theta'(p)^2}{48}\partial_x^2\rho_{x-v(p)t,0}(-p) + \right.$$
$$\left. \frac{3\theta''(p)^2 - 4\theta'(p)\theta'''(p) - 2\theta'(p)^4}{768}\partial_x^4\rho_{x-v(p)t,0}(-p)\right). \quad (128)$$

This means

$$(\partial_t + v(p)\partial_x)\rho_{x,t}(p) = \frac{\frac{v''(p)}{24}\partial_x^3 - \frac{v^{iv}(p)}{1920}\partial_x^5 + \frac{tv''(p)^2}{576}\partial_x^6}{1 + \frac{t}{24}v''(p)\partial_x^3 - \frac{t}{1920}v^{iv}(p)\partial_x^5 + \frac{t^2}{1152}v''(p)^2\partial_x^6}\rho_{x,t}(p) \approx$$
$$\frac{v''(p)}{24}\partial_x^3\rho_{x,t}(p) - \frac{v^{iv}(p)}{1920}\partial_x^5\rho_{x,t}(p). \quad (129)$$

## B.2 Off-diagonal states

An off-diagonal state is completely characterised by the equations

$$\sum_{n=0}^m \frac{(-i)^n}{n!} \sum_{s=\pm 1} \partial_q^n \left\{ \frac{1+s\sigma^y e^{i\vec{\theta}\left(p+\frac{q}{2}\right)\cdot\vec{\sigma}}}{2} \times \right.$$
$$\left. \hat{\Gamma}_{x-st\frac{\varepsilon\left(sp+s\frac{q}{2}\right)-\varepsilon\left(sp-s\frac{q}{2}\right)}{q}}^{(m-n)}(e^{ip}) \frac{1+s\sigma^y e^{i\vec{\theta}\left(p-\frac{q}{2}\right)\cdot\vec{\sigma}}}{2} \right\}_{q=0} = 0. \quad (130)$$

By truncating the number of equations to $m = N$, the error is $O\left(1/\chi^{N+1}\right)$. A single odd complex field $\Psi_x(p)$ completely characterises an off-diagonal state; the first terms of the expansion of its symbol are the following

$$\Gamma_x^{(0)}(p) = 4\pi\left[\Psi_x(p)\frac{\sigma^z + i\sigma^x e^{i\theta\sigma^z}}{2} + h.c.\right]$$
$$\Gamma_x^{(1)}(p) = -\pi\theta'(p)I[\Psi_x(p) + \Psi_x^*(p)]$$
$$\Gamma_x^{(2)}(p) = \pi\frac{\theta''(p)}{4}\sigma^y e^{i\theta(p)\sigma^z}i[\Psi_x^*(p) - \Psi_x(p)]$$
$$\Gamma_x^{(3)}(p) = \pi\frac{\theta'''(p) + 2\theta'(p)^3}{24}I[\Psi_x(p) + \Psi_x^*(p)] \quad (131)$$
$$\Gamma_x^{(4)}(p) = -\pi\frac{\theta''''(p)}{192}\sigma^y e^{i\theta(p)\sigma^z}i[\Psi_x^*(p) - \Psi_x(p)]$$
$$\Gamma_x^{(5)}(p) = -\pi\frac{16\theta'(p)^5 + 20\theta'(p)^2\theta'''(p) + \theta^v(p)}{1920}I[\Psi_x(p) + \Psi_x^*(p)].$$

We have defined $\Psi_x(p)$ in such a way that, at the leading order, it corresponds to the bare field $\Psi_x^{bare}(p)$ as defined in (21).

**Time evolution.** The symbol of an off-diagonal state time evolves as follows:

$$\hat{\Gamma}_{x,t}^{(m)}(e^{ip}) = \sum_{n=0}^{m} \frac{(-i)^n}{n!} \sum_{s=\pm 1} e^{ist\varepsilon(-sp)+\varepsilon(sp)]} \partial_q^n \left\{ \frac{1 - s\sigma^y e^{i\vec{\theta}\left(p+\frac{q}{2}\right)\cdot\vec{\sigma}}}{2} \times \right.$$

$$\left. \hat{\Gamma}_{x-st\frac{\varepsilon(-sp)+\varepsilon(sp)-\varepsilon\left(-sp-s\frac{q}{2}\right)-\varepsilon\left(sp-s\frac{q}{2}\right)}{q},0}^{(m-n)}(e^{ip}) \frac{1 + s\sigma^y e^{i\vec{\theta}\left(p-\frac{q}{2}\right)\cdot\vec{\sigma}}}{2} \right\}_{q=0}, \quad (132)$$

where $\hat{\Gamma}_{x,t}^{(n)}(e^{ip})$ are written as in (131).

Up to $O\left(1/\xi^3\right)$ corrections, the complex field time evolves as follows:

$$\Psi_{x,t}(p) = e^{-it[\varepsilon(p)+\varepsilon(-p)]} \left( 1 + \frac{it[v'(p)+v'(-p)]}{8} \partial_x^2 + \frac{t[v''(p)+v''(-p)]}{48} \partial_x^3 - \right.$$

$$\left. \frac{t^2[v'(p)+v'(-p)]^2}{128} \partial_x^4 \right) \left( \Psi_{x-t\frac{v(p)+v(-p)}{2},0}(p) - \frac{\theta'(p)^2}{16} \partial_x^2 \Psi^*_{x-t\frac{v(p)+v(-p)}{2},0}(p) \right). \quad (133)$$

By taking the derivative with respect to the time we obtain

$$\left( \partial_t + \frac{v(p)+v(-p)}{2} \partial_x \right) \left[ e^{it[\varepsilon(p)+\varepsilon(-p)]} \Psi_{x,t}(p) \right] =$$

$$\frac{\frac{i[v'(p)+v'(-p)]}{8}\partial_x^2 + \frac{[v''(p)+v''(-p)]}{48}\partial_x^3 - \frac{t[v'(p)+v'(-p)]^2}{64}\partial_x^4}{1 + \frac{it[v'(p)+v'(-p)]}{8}\partial_x^2 + \frac{t[v''(p)+v''(-p)]}{48}\partial_x^3 - \frac{t^2[v'(p)+v'(-p)]^2}{128}\partial_x^4} \left[ e^{it[\varepsilon(p)+\varepsilon(-p)]}\Psi_{x,t}(p) \right] \approx$$

$$\frac{i}{4} \frac{v'(p)+v'(-p)}{2} \partial_x^2 \left[ e^{it[\varepsilon(p)+\varepsilon(-p)]}\Psi_{x,t}(p) \right] +$$

$$+ \frac{1}{24} \frac{v''(p)+v''(-p)}{2} \partial_x^3 \left[ e^{it[\varepsilon(p)+\varepsilon(-p)]}\Psi_{x,t}(p) \right] \quad (134)$$

and hence

$$i\partial_t \Psi_{x,t}(p) = 2\frac{\varepsilon(p)+\varepsilon(-p)}{2}\Psi_{x,t}(p) - i\frac{v(p)+v(-p)}{2}\partial_x \Psi_{x,t}(p) - \frac{1}{4}\frac{v'(p)+v'(-p)}{2}\partial_x^2\Psi_{x,t}(p) +$$

$$\frac{i}{24}\frac{v''(p)+v''(-p)}{2}\partial_x^3\Psi_{x,t}(p) + O\left(\xi^{-3}\right). \quad (135)$$

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
