# Peer review of "Locally quasi-stationary states in noninteracting spin chains"

_SciPost Physics, doi:SciPost Phys. 8, 048 (2020)_

## Round 2 · Referee Report · Olalla Castro-Alvaredo (Referee 1) · 2019-11-14

Strengths

1) It addresses several problems of current interest in this area of research. 2) It proposes a novel solution in the form of a new Mathematical framework to treat a variety of models and out-of-equilibrium situations. 3) It is very well written. 4) The solution proposed in particular at the level of (generalized) GHD equations is very elegant and natural. In particular the use of Moyal products is a very interesting feature that might suggest a way to further generalizations.

Weaknesses

1) The paper is very technical so not particularly easy to read but as clear as possible given the nature of the results. 2) The techniques presented only apply to free theories.

Report

In my view this paper contains (at least) two important results: the introduction of an unambiguous definition of the charge densities and associated currents, by introducing precisely defined locally quasi-conserved operators, and a hydrodynamic description of the dynamics of a large class of non-interacting theories by means of (what could be called!) a generalized GHD.

I find the latter result most impressive. The proposed generalization captures all higher order (quantum) corrections to the known GHD equations within an extremely elegant, and seemingly very natural, mathematical framework. “Standard” products are replaced by carefully defined Moyal products, a solution that is very satisfying given that this replacement is supposed to generate all quantum corrections to GHD.

The paper is very technical and therefore not very easy to read but I find that it is well written, and as clear as possible given the nature of the results.

Among the many papers currently being produced in relation to different applications/generalizations of hydrodynamics and find this an impressive and original contribution. Beyond specific models and quenches, it proposes a new mathematical framework to treat a large variety of problems of current interest and an elegant solution to a family of such problems within that framework.

Of course, the results are not fully model-independent (as they only apply to free theories) and the big question is how this might be generalized to interacting systems, as acknowledged in the paper itself. I am sure there will be subsequent work on this. For the moment I think this is a very good contribution to the field and should certainly be published in SciPost.

I have only small comments to make as I found the paper very good overall.

Requested changes

1) I have a small comment/question regarding the statement at the top of page 3 “incorrect definition of space dependent root densities”.

I think I understand what is meant by this, specially given Section 3, however reference [35] is also cited at this point and I do not think this statement can apply to the QFT case. Certainly “root densities” do not play a role there (particle densities do), but perhaps this “incorrect definition” translates into some other issue in the QFT framework. Could the author clarify this a little bit? Or remove the reference at that stage, if the statement does not apply?

2) In general, I find the paragraph containing this sentence and the one before it, in the previous page, a little bit too negative about GHD. First, GHD was formulated to describe the Euler scale and at that scale none of the problems mentioned here arise, so there is no “incorrect definition of the roots densities” in that original set-up, they are perfectly well defined within that framework. Second, the treatment of diffusive phenomena is referred to as if it were a completely separate development from GHD. In my view it is rather an extension of GHD beyond Euler scale. It is not the complete extension that is presented in this paper, but it is a further refinement of GHD that now includes diffusive phenomena. And within the framework of diffusion, particularly the works [78,79], the “ambiguity” problem raised in this paper is dealt with by fixing a particular gauge, the PT-invariant gauge. Within that gauge, as far as I understand, their results are fully well defined. I think this deserves a mention in this introductory section and maybe in section 3 too, in particular when the issue of gauge fixing is dealt with.

3) I also found the last sentence before section 1.2 a bit obscure. Would it be a possible to say something a bit more concrete about these possible “unfortunate conventions” and how they might generate “cumbersome” large time corrections? Are there existing examples of this in the literature that he could cite?

  • validity: top
  • significance: top
  • originality: top
  • clarity: high
  • formatting: excellent
  • grammar: excellent

Author:  Maurizio Fagotti  on 2019-11-21  [id 652]

(in reply to Report 1 by Olalla Castro-Alvaredo on 2019-11-14)
Category:
remark
answer to question

I am glad that the referee has a very positive view of my work, and I thank her/him for having read the manuscript.
Before resubmitting a new version, I would like to answer the main points raised by the referee.

The first two comments have made me realise that a definition can not be incorrect, but it can be inappropriate. To the best of my knowledge, in integrable systems with translationally invariant Hamiltonians the root densities describe a class of homogeneous stationary states, therefore the current definition of a space-dependent root density in interacting systems is not incorrect but missing.
On the other hand, in [doi:10.1103/PhysRevB.96.220302] I had provided a definition of space-dependent root densities in noninteracting systems. Unfortunately, that definition is not appropriate to describe models with a nontrivial Bogoliubov angle, as the so-defined root densities do not evolve in a decoupled way; as a consequence, generalised hydrodynamics remains correct only at the Euler scale. In this new paper I have showed that a redefinition of the root densities is sufficient to fix that problem, making GHD an exact theory.

About the "ambiguity" problem, the gauge fixing of Refs [78,79] has a completely different goal. As far as I currently understand, the authors needed to fix the gauge because they were investigating a gauge-dependent observable; to that aim, they identified a symmetry that allows for an unambiguous definition of the charge densities. In my work the gauge can not be fixed in an arbitrary way: a gauge change is the only tool to isolate the invariant subspace. In other words, I am addressing a problem that has been almost always overlooked (in the context of GHD), namely, that an inappropriate definition of space-dependent root density results in the impossibility to write a dynamical equation (a higher order GHD equation) written solely in terms of root densities.

Concerning the third comment, an example of ``cumbersome'' large time corrections is provided again by [doi:10.1103/PhysRevB.96.220302], where the inappropriate definition of root densities, and, in turn, of higher-order GHD, did not allow me to capture a universal part of the light-cone dynamics (and, in turn, to reproduce the numerical data).

---

## Round 2 · Referee Report · Anonymous (Referee 2) · 2020-1-16

Strengths

1- the paper addresses a very timely subject as the systematic computation of corrections to GHD 2- the developed framework is completely general for noninteracting theories and elegant 3- generalizations to the interacting case can be envisaged, although this can be hard in practice

Weaknesses

1- the paper is very technical and to maintain its degree of generality, it is based on a notation which is not known to everybody 2- many references are made to previous papers by the same author, which makes the current manuscript less self-contained 3- there is no quantitative numerical verification of the higher corrections for GHD. This could be useful for Sec. 4 for instance.

Report

This paper considers the problem of developing a complete theory for the inhomogeneous dynamics of 1d noninteracting fermionic systems. At the lowest order in the inhomogeneity, GHD provides the full answer but several efforts have been recently made to characterize higher order corrections. This paper takes a step back which is important and useful in general: to provide unambiguous definitions of the quantities which GHD (and its higher order corrections) tries to describe (e.g. root densities, etc). At least, in the noninteracting case, this work provides an elegant approach, which by making use of the available gauge freedom, identifies a whole invariant subspace of conserved charges, currents, currents of currents, etc.
This results in Eq. (44) and (45) which compactly represent the main result of the paper.
This is for sure a relevant and interesting paper that deserves publication. The main issue I can see is that the paper is rather technical, makes use of a particular notation which is not easy to grasp, and requires knowledge of more than one paper from the same author. So it is not easy to read even for someone in the same field.

I understand that for most of its content, technicalities are practically unavoidable. However, I think it would be useful if the introduction of the paper was expanded in order to give a schematic intuition of its content and main results. For instance, a brief recap of the problems encountered in [83] which this new manuscript is solving, would certainly help.

Requested changes

1- In Eq. (4) the definition of G is not given. 2- On page 4, can the author comment on why is it obvious that Eq. (5) is quasilocal for C[Q] = 0 for noninteracting systems? 3-Page 7: the notion of "physical part" is not entirely clear. Can the author provide an example for a simple case (e.g. the Ising spin chain) 4- I had some troubles understanding the definition of "global" / "local" root density and auxiliary complex field. The quantities introduced in Eq. 23 are global but do depend on the space index x. Some additional comments would be useful and could help understanding the Localisation procedure performed in page 14.

  • validity: top
  • significance: high
  • originality: top
  • clarity: ok
  • formatting: excellent
  • grammar: excellent

Author:  Maurizio Fagotti  on 2020-02-24  [id 745]

(in reply to Report 2 on 2020-01-16)
Category:
remark
answer to question

I thank the referee for having read the manuscript and for pointing out potential obstacles to its reading. I have taken into account the criticisms that I found appropriate, and I think that the revised version of the manuscript is a substantial improvement on the original version. First, I would like to comment on the weaknesses spotted by the referee. W1: "the paper is very technical and to maintain its degree of generality, it is based on a notation which is not known to everybody". I agree that the paper is technical, but I do not see how I could have addressed the problem differently. I do not agree, however, that the notation is unknown. A reader familiar with (block-)Toeplitz matrices/operators knows what a "symbol" is. A reader unfamiliar with such structured matrices can take advantage of section 2, where I have introduced all the required elements to understand the paper. As a matter of fact, I've made a big effort to use standard notations and to establish connections with the quantities used in the literature. In the end, everything is expressed in terms of "root densities", "excitation energies", "single-particle eigenvalues", etc., which are the quantities usually used in the field. W2: "many references are made to previous papers by the same author, which makes the current manuscript less self-contained". Maybe the referee was confused by a sentence that I wrote at the end of section 2, where I referred to a review of mine for additional details. This paper has been conceived to be self-contained; I read the paper again and I have been unable to find references to papers that must be read in order to understand the manuscript. The bibliography has more than 100 references and I have only cited the most relevant papers of mine. W3: "there is no quantitative numerical verification of the higher corrections for GHD. This could be useful for Sec. 4 for instance". I agree with the referee that the original version had this weakness. I resolve this issue in the revised version.

My answers to the requested changes follow. 1. The quantity G will be explicitly defined (in fact, it was implicitly defined by its density even in the original version). 2. The paragraph will be rephrased in such a way that also a reader unfamiliar with the formalism will not feel disoriented. 3. I called "unphysical" the Fourier components of the symbol that do not affect the operators and the correlation matrices. The "physical part" of the symbol is, in turn, the part that can not be arbitrarily changed. A sentence will be rephrased to avoid similar doubt. 4. I thank the referee for pointing out that the term "global" is misleading. I will replace "global root density" by "spuriously local root density". Such quantity depends indeed on the position but can not be considered a genuinely local root density because time evolution mixes it with independent degrees of freedom.

---

## Round 3 · Author Response

The manuscript has been improved following the referees' comments; in addition, some parts have been refined. The main changes are in sections 1.1, 3.4, and 4.
In section 1.1 I've clarified the problems that were undermining generalised hydrodynamics, explaining, in particular, the issues found in Ref. [84]. I've also commented on the gauge choice of Refs [79][80].
In the original version of the manuscript, the solution to the dynamics in the presence of Hamiltonian inhomogeneities was arguably unsatisfactory; I resolved this issue by rewriting part of section 3.4 (section 3.3 of the original version).
Section 4 has been expanded to include a numerical check of the theory. Specifically, I used a trick to adapt the results of the manuscript to finite chains with periodic boundary conditions on the Jordan-Wigner fermions; this has allowed me to numerically test the predictions in finite chains, finding perfect agreement. I've also shown the error made by truncating generalised hydrodynamics at a given order.
Overall, I've taken into consideration the referees' recommendations and I've also made changes to resolve the weaknesses pointed out by referee 2. I think that the new version of the manuscript is a substantial improvement on the original version, and I am confident that the referees will appreciate the changes.
In section 1.1 I've clarified the problems that were undermining generalised hydrodynamics, explaining, in particular, the issues found in Ref. [84]. I've also commented on the gauge choice of Refs [79][80].
In the original version of the manuscript, the solution to the dynamics in the presence of Hamiltonian inhomogeneities was arguably unsatisfactory; I resolved this issue by rewriting part of section 3.4 (section 3.3 of the original version).
Section 4 has been expanded to include a numerical check of the theory. Specifically, I used a trick to adapt the results of the manuscript to finite chains with periodic boundary conditions on the Jordan-Wigner fermions; this has allowed me to numerically test the predictions in finite chains, finding perfect agreement. I've also shown the error made by truncating generalised hydrodynamics at a given order.
Overall, I've taken into consideration the referees' recommendations and I've also made changes to resolve the weaknesses pointed out by referee 2. I think that the new version of the manuscript is a substantial improvement on the original version, and I am confident that the referees will appreciate the changes.

---

## Round 3 · List of Changes

- Some typos have been fixed.
- Section 1.1 has been improved.
- To avoid misunderstanding, in section 2 some sentences have been rephrased.
- The terminology "global/local root density" and "global/local auxiliary field" has been revised. The quantities are now called "spuriously local" instead of "global" and "genuinely local" instead of "local".
- For the sake of simplicity, some irrelevant comments in Section 2 have been removed.
- A short subsection called "Expectation values" has been added after 3.2; it contains some pre-existent material of the original version (section 3.2) and something new.
- Section 3.4 (i.e. section 3.3 of the original version) has been substantially improved.
- A subsection has been created in section 4. It reports a numerical check of the analytical predictions.
- Three references have been added: [88], [89], and [91].

---

## Editorial Decision

published